# FedPAC: Consistent Representation Learning for Federated Unsupervised Learning under Data Heterogeneity

## Abstract

Federated unsupervised learning enables collaborative model training on decentralized unlabeled data but faces critical challenges under data heterogeneity, which often leads to representation collapse from weak supervisory signals and semantic misalignment across clients. Without a consistent semantic structure constraint, local models learn disparate feature spaces, and conventional parameter averaging fails to produce a coherent global model. To address these issues, we propose Federated unsupervised learning with Prototype Anchored Consensus (FedPAC), a novel framework that establishes a consistent representation space via a set of learnable prototypes. FedPAC introduces a dual-alignment objective during local training: a semantic alignment loss that steers local models towards a prototype-anchored consensus to ensure cross-client semantic consistency, coupled with a representation alignment loss that promotes the learning of discriminative and stable features. On the server, prototypes are aggregated by an optimization-based strategy that preserves semantic knowledge and ensures the prototypes remain representative. We provide a rigorous convergence analysis for our method, formally proving its convergence under mild assumptions. Extensive experiments on benchmarks including CIFAR-10 and CIFAR-100 demonstrate that FedPAC significantly outperforms state-of-the-art methods across a wide range of heterogeneous settings.

## 1 Introduction

Federated learning (FL) (McMahan et al., 2017) enables a set of distributed clients to collaboratively train a shared model without exchanging raw data, thereby providing privacy preservation. A central challenge in FL is data heterogeneity: clients typically hold non-IID local datasets, and such distributional skew ca cause local updates to conflict, degrade the aggregated global model, and destabilize convergence. Existing FL algorithms generally assume supervised local training with abundant, high-quality labels. However, it is often impractical to collect large-scale, accurately annotated datasets in many practical applications. This pervasive label scarcity not only limits attainable performance but also undermines generalization to new domains, motivating methods that exploit the large volumes of unlabeled data distributed across clients. In this work, we study representation learning for federated unsupervised learning with non-IID data (Jin et al., 2023), aiming to extract robust and generalizable representations from distributed, unlabeled, and imbalanced data.

Federated unsupervised learning currently faces two fundamental challenges that impede the training of a high-quality global model. The first is **representation collapse**, where the weak supervisory signals from unlabeled data can lead to degenerate features with limited discriminability. Data heterogeneity further exacerbates training instability, increasing the risk of representation collapse. The second, and more complex challenge is **cross-client semantic misalignment**. Data heterogeneity across clients undermines the objective of learning a unified global representation, as each client learns distinct feature spaces tailored to its local data distribution. This causes representations of semantically similar samples to drift to disparate regions of the global feature space. This misalignment is often exacerbated by simple parameter averaging, which can blur semantic boundaries and paradoxically degrade the performance of the global model. These issues expose a fundamental tension, i.e., how to learn representations that are both locally discriminative and globally coherent.

Several prior works have attempted to apply self-supervised learning (SSL) methods that have proven effective in centralized settings, e.g., SimCLR (Chen et al., 2020), BYOL (Grill et al., 2020), and SimSiam (Chen & He, 2021), to client-side local training within FL frameworks. However, these approaches often rely on large batch sizes or extensive negative samples, which are not applicable in resource-constrained FL environments. Crucially, as client heterogeneity increases, straightforward extensions of centralized SSL methods to FL scenarios often results in degraded performance. Alternative strategies have been proposed, including aggregating models via knowledge distillation (Han et al., 2022), local clustering (Lubana et al., 2022), and promoting unified representation by constraining consistent client model updating (Liao et al., 2024). While these methods have made partial strides, they primarily focus on preventing local representation collapse and lack explicit mechanisms to enforce semantic consistency across clients, rendering them vulnerable to representation drift under data heterogeneity. No existing approach adequately addresses both representation collapse and semantic misalignment in a unified manner so far, highlighting the need for a more principled approach to semantic-aware federated representation learning.

To bridge this critical gap, we propose Federated Unsupervised Learning with Prototype Anchored Consensus (FedPAC), a framework that resolves the tension between local learning and global consistency through a set of globally shared, learnable prototypes. On the client-side, we introduce a dual-alignment learning objective. At the representation level, we leverage self-supervised learning to promote discriminative feature learning and prevent collapse. At the semantic level, each client aligns local features to the prototypes, ensuring that representations corresponding to similar concepts are mapped to a globally consistent representation space regardless of local data distribution, significantly mitigating representation drift. On the server-side, we design a prototype aggregation strategy that refines the global prototypes by integrating semantic insights from clients, ensuring the prototypes remain diverse and globally representative throughout training. Through the interplay of local dual-alignment and server aggregation, FedPAC learns a unified representation space that is both locally discriminative and globally coherent, overcoming the limitations of prior federated unsupervised learning methods.

In summary, in this work we propose FedPAC to tackle the key challenges of representation collapse and counteracts semantic misalignment in federated unsupervised learning. The core of our approach is a prototype-based semantic anchoring mechanism that establishes a globally consistent feature space across clients under non-IID data. (1) We propose a synergistic architecture that combines a dual-alignment learning objective for clients' local unsupervised learning and a prototype aggregation strategy refining global prototypes on the server. (2) We provide a rigorous convergence analysis that theoretically establishes the stability and soundness of our proposed method. (3) Empirically, we validate FedPAC through extensive experiments on two benchmark datasets. The results show that our framework significantly outperforms state-of-the-art methods, confirming the practical effectiveness of our semantic alignment strategy.

## 2 RELATED WORK

### 2.1 SELF-SUPERVISED LEARNING

SSL has advanced rapidly in recent years, enabling the learning of transferable representations without manual annotation. Current SSL methods are broadly categorized into discriminative and predictive approaches. Discriminative methods learn representations by enforcing invariance at the instance or cluster level. While effective in centralized settings, these methods face significant challenges under the constraints of federated learning. Contrastive learning, e.g., SimCLR (Chen et al., 2020), MoCo (He et al., 2020), are hampered by their reliance on large batch sizes or substantial negative samples, which are impractical on resource-constrained clients. Non-contrastive bootstrap methods like BYOL (Grill et al., 2020) and SimSiam (Chen & He, 2021), which typically depend on batch statistics for normalization and stabilization, are sensitive to data heterogeneity and can exacerbate client drift. Similarly, clustering approaches such as DeepCluster (Caron et al., 2018) and SwAV (Caron et al., 2020) often impose equipartition constraints to prevent collapse, which is ineffective under the imbalanced class distributions of non-IID data. Predictive methods, which learn through reconstruction (He et al., 2022) or pretext tasks (Gidaris et al., 2018), are less suitable for federated settings due to their high computational and communication costs.

## 2.2 Federated Unsupervised Learning

Recent research has begun to address the challenges of federated unsupervised learning, focusing on mitigating data heterogeneity and learning consistent representations across clients. A common strategy combines local SSL method with specialized alignment mechanism. For instance, FedCA (Zhang et al., 2023) and ProtoFL (Kim et al., 2023) employ a shared dictionary and prototypical distillation, respectively, to maintain the consistency of representation. FedX (Han et al., 2022) further enhances this via bidirectional distillation. To address divergence caused by non-IID data, researchers have optimized the aggregation process. FedU (Zhuang et al., 2021) and FedEMA (Zhuang et al., 2022) adjust predictor aggregation modules and EMA decay adaptive to the divergence. Similarly, L-DAWA (Rehman et al., 2023) mitigates the impact of client bias during aggregation according to the divergence between local and global models. Orchestra (Lubana et al., 2022) uses local clustering tasks to learn representations and coordinates them through a hierarchical structure to enforce globally consistent cluster assignments. More recently, attention has shifted towards preventing representation collapse and optimizing feature space structure. FedU2 (Liao et al., 2024) encourages uniform distribution of representations, while FedDecorr (Shi et al., 2023) introduces a regularization term to decorrelate dimensions and prevent dimensional collapse. Beyond these, SSD (Fang et al., 2025) introduces a soft separation mechanism via dimension-scaled regularization to enhance inter-client uniformity. Despite these advances in structural alignment, ensuring semantic consistency across heterogeneous clients remains a challenge.

## 2.3 Optimal Transport

Optimal Transport (OT) (Villani et al., 2008) provides a framework for aligning probability distributions by minimizing displacement costs. It has gained prominence in machine learning for tasks like domain adaptation, generative modeling (Courty et al., 2017; Tolstikhin et al., 2017; Balaji et al., 2020) and self-supervised learning methods like SwAV (Caron et al., 2020), which utilizes OT for assignment to perform online clustering and enable self-supervised learning without contrastive pairs. Recently, this paradigm has extended to graph learning, where OT aligns complex structural representations across divergent domains (Deng et al., 2025; Fu et al., 2025). However, standard OT enforces strict mass conservation, which can be restrictive in the presence of noise or outliers. To address this, Unbalanced Optimal Transport (UOT) (Benamou, 2003; Séjourné et al., 2019) relaxes marginal constraints using divergence penalties, offering a more robust solution for such scenarios (Liu et al., 2023; Wang et al., 2024).

## 3 Preliminaries

Before detailing our methods, we introduce necessary definitions here. We also present Table 3 in Appendix A.2, providing a comprehensive explanation of the notations used throughout this paper.

We formulate Federated Unsupervised Learning (FUL) as follows. Consider a federated system consisting of a central server and $N$ clients. Each client $n$ holds a local unlabeled dataset $D_n = \{x_{n,i}\}_{i=1}^{|D_n|}$, where $|D_n|$ is the number of samples on client $n$. In practice, the data distributions across different clients are non-IID. A standard FUL problem can be formulated as a distributed optimization, aiming to collaboratively learn a global model $\theta$ across $N$ clients, i.e.,

$$\min_\theta F(\theta) = \sum_{n=1}^{N} w_n F_n(\theta), \quad \text{with} \quad F_n(\theta) = \mathbb{E}_{x \sim D_n}\big[\mathcal{L}_n(\theta; x)\big], \quad (1)$$

where $w_n$ denotes the aggregation weight of client $n$ satisfying $w_n \geq 0$ and $\sum_n w_n = 1$, and $F_n(\theta)$ is the local objective based on client-specific unsupervised loss $L_n(\cdot)$.

For local unsupervised training, we adopt the online-target network design similar to BYOL (Grill et al., 2020). The online network, parameterized by $\theta$, decomposes into an encoder $f_\theta$, a projector $h_\theta$ and a predictor $q_\theta$. For an input $x$ the online network outputs representation $\mathbf{y} = f_\theta(x)$, projection $\mathbf{z} = g_\theta(\mathbf{y})$ and prediction $q_\theta(\mathbf{z})$. The target network, parameterized by $\phi$, is an exponential moving average of the online encoder and projector that receives no gradients and provides stable targets for the online predictor to match. Our method also relies on learning invariance to data augmentations.

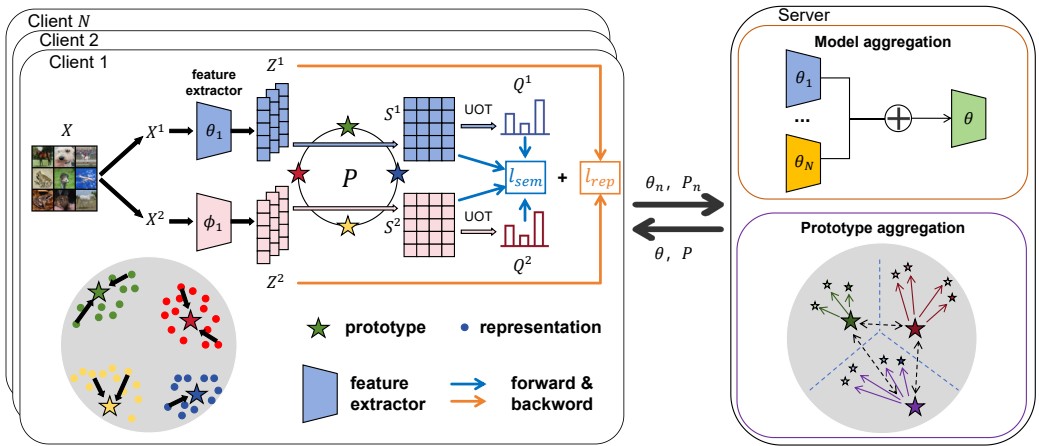

Figure 1: Overview of the FedPAC framework. On the client side, local learning is driven by both representation and semantic alignment loss. On the server side, model parameters are aggregated via Federated Averaging, while global prototypes are refined using an optimization-based strategy to enhance semantic consensus across the federation.

We define a stochastic augmentation function $\mathcal{T}$ that transforms an input $x$ by randomly sampling $t' \sim \mathcal{T}$ to produce the augmented view $x' = t(x)$. In our setting, each client applies the same augmentation pipeline $\mathcal{T}$ independently to ensure consistency of augmented views across clients.

# 4 METHOD

## 4.1 OVERVIEW

To address the challenges of representation learning in federated unsupervised learning under non-IID data, we propose FedPAC. Our framework introduces two complementary objectives into the client-side unsupervised training, i.e., (1) a representation alignment loss that promotes the learning of view-invariant and discriminative features, thereby ensuring robustness against collapse, and (2) a semantic alignment loss that aligns local representations with a set of globally shared prototypes, thus enforcing a consistent semantic structure across the entire federation. This dual-pronged strategy yields a unified feature space that is both locally discriminative and globally consistent.

The overall training pipeline of FedPAC is illustrated in Figure 1. At the beginning of each communication round, the server broadcasts the current global model parameters and the global prototypes to the participating clients. Each selected client then perform $E$ epochs of local unsupervised training by minimizing the local objective $\mathcal{L}_{\text{local}}$, which simultaneously optimizes its model parameters and local prototypes. Upon completion of local training, clients send their updated model parameters and local prototypes to the server for aggregation. Through iterating this process, the global model and prototypes are jointly refined, learning an effective and consistent representation space from distributed, unlabeled, and imbalanced data.

## 4.2 CLIENT-SIDE UNSUPERVISED TRAINING

When performing local unsupervised training, each client is designed to learn representations that are both discriminative and semantically consistent. For each batch, two augmented views of each sample are generated and processed by the feature extractor. We then compute the prototype assignments via optimal transport, predicated on the similarity between projections and prototypes, and yields the corresponding semantic alignment loss $\ell_{\text{sem}}$. Concurrently, a representation-alignment loss $\ell_{\text{rep}}$ is employed to align representations between views, supplemented by a rotation prediction task to enhance stability in early training. The comprehensive local objective is formulated as $\mathcal{L}_{\text{local}} = \ell_{\text{rep}} + \lambda\,\ell_{\text{sem}}$, minimized via SGD to jointly optimize model parameters and prototypes, with $\lambda$ balancing the two terms. We detail the design of these two loss components below.

### 4.2.1 PROTOTYPE-BASED SEMANTIC ALIGNMENT

To facilitate semantic alignment across clients, we introduce a set of learnable prototypes $\{\mathbf{p}_k\}_{k=1}^{K}$ that serve as shared semantic anchors. Rather than directly aligning raw feature spaces, clients learn by mapping their local representations to a probability distribution over these shared prototypes. By enforcing consistency in these distributions for similar semantics across different views and clients, we achieve cross-view and cross-client semantic alignment anchored by the global prototypes.

**Computing prototype assignments under heterogeneity.** Given a batch of projections $\mathbf{Z} \in \mathbb{R}^{B \times d}$ and the prototypes $\mathbf{P} \in \mathbb{R}^{K \times d}$, our objective is to compute the prototype assignment matrix $\mathbf{Q} \in \mathbb{R}^{B \times K}$, where $\mathbf{Q}_{i,k}$ represents the probability mass assigned from sample $i$ to prototype $k$. This both mitigates trivial collapse that arises from assigning each sample to its nearest prototype and provides a smooth training target for the cross-view prediction loss. We determine the optimal $\mathbf{Q}$ by solving an optimal transport problem.

In heterogeneous settings, clients often possess highly imbalanced data distributions. While enforcing a uniform distribution over prototype selections, as done in some clustering methods, is effective at preventing collapse, it can lead to inappropriate assignments and consequently degrade the quality of the learned representation. To address this challenge, we employ Unbalanced Optimal Transport (UOT) to compute the soft assignments. UOT allows marginals to deviate from strictly uniform distribution, thereby better accommodating scenarios where clients lack certain classes. In our setting we employ a semi-relaxed UOT variant that replaces hard marginal equalities on prototypes with KL penalties while retaining an entropic regularizer, i.e.,

$$\min_{\mathbf{Q} \in \mathbb{R}_+^{B \times K}} \mathrm{Tr}\big(-\mathbf{Q}^\top \mathbf{Z} \mathbf{P}^\top\big) - \varepsilon H(\mathbf{Q}) + \mu \, \mathrm{KL}\big(\mathbf{Q}^\top \mathbf{1}_B \,\|\, \frac{1}{K}\mathbf{1}_K\big) \quad s.t. \quad \mathbf{Q}\mathbf{1}_K = \frac{1}{B}\mathbf{1}_B. \tag{2}$$

Here, $H(\mathbf{Q}) = -\sum_{i,k} \mathbf{Q}_{i,k}$ and the entropic parameter $\varepsilon$ controls the smoothness of the assignment. $\mathrm{KL}(\cdot\|\cdot)$ is the Kullback–Leibler divergence penalizing the deviation of prototype marginal $\mathbf{Q}^\top \mathbf{1}_B \in \mathbb{R}^K$ from uniform distribution. A hard uniform constraint is enforced on sample marginal $\mathbf{Q}\mathbf{1}_K$ to ensure equal assignment across samples, while the KL penalty with strength parameter $\mu \geq 0$ softly encourages balanced prototype usage. This asymmetric constraint design prevents trivial collapse while allowing prototype marginals to deviate from strict uniformity, thereby producing more reliable assignments under non-IID data and still promoting balanced prototype utilization. This formulation can be efficiently solved using generalized Sinkhorn iterations, and we provide the detailed derivation in Appendix A.1.

**Computing Semantic Alignment Loss.** For each batch of samples, we generate two augmented views, yielding projections $\mathbf{Z}^{(1)}$ and $\mathbf{Z}^{(2)}$. We first compute the optimal soft assignment $\mathbf{Q}^{(1)}$ from $\mathbf{Z}^{(1)}$ using the above procedure, and similarly $\mathbf{Q}^{(2)}$ from $\mathbf{Z}^{(2)}$. Then we require the projections obtained from one view to predict the assignment of the other. Since projections lack interaction with the prototypes, we compute a similarity probability matrix and the semantic alignment loss is defined as the sum of cross-entropy between assignment and similarity in both directions, i.e.,

$$\ell_{\mathrm{sem}} = -\frac{1}{2B} \sum_{i=1}^{B} \sum_{k=1}^{K} \Big( \mathbf{Q}_{i,k}^{(1)} \log \mathbf{S}_{i,k}^{(2)} + \mathbf{Q}_{i,k}^{(2)} \log \mathbf{S}_{i,k}^{(1)} \Big), \tag{3}$$

where $\mathbf{S}_{i,k} = \frac{\exp(\mathbf{z}_i^\top \mathbf{p}_k / \tau)}{\sum_{k'} \exp(\mathbf{z}_i^\top \mathbf{p}_{k'} / \tau)}$ with $\tau$ controlling the sharpness (Wu et al., 2018; Caron et al., 2020). Minimizing this loss encourages projections from different views of the same image to share the same prototype assignment. This objective jointly optimizes both representations and prototypes, simultaneously pulling each feature towards its assigned prototypes while also moving each prototype towards the centroid of its assigned features. Through this co-optimization, model structures the feature space into semantically distinct clusters anchored by prototypes, thereby promoting both view invariance and cross-client semantic alignment even under significant data heterogeneity.

### 4.2.2 CONTRASTIVE REPRESENTATION ALIGNMENT AND STABILIZATION

In federated unsupervised learning, the lack of ground truth labels often leads to weak supervisory signals, increasing the risk of representation collapse. Self-supervised learning mitigates this by enforcing augmentation invariance to promote compact and well-separated feature clusters. Integrated with our proposed semantic alignment loss, SSL methods not only align representations across different views to stabilize local training, but also yield discriminative features that facilitate semantic alignment. In our work, we adopt a similar architecture as BYOL for its stable and negative-free learning signal, which is well suited to resource-constrained FL environments. However, during the initial stages of training, prototypes may correspond to random or weakly discriminative features and thus provide unstable supervisory signals. Inspired by (Lubana et al., 2022) and other works that add inexpensive predictive tasks to stabilize early training, we introduce a rotation prediction task to encourage the formation of stable and meaningful representations before prototype stabilization.

Concretely, for each sample $x$ we generate two augmented views $x^{(1)}, x^{(2)} \sim \mathcal{T}(x)$ along with rotated versions $\tilde{x}$. Rotation angles are randomly sampled from $\{0°, 90°, 180°, 270°\}$, with corresponding label $\varphi \in \{0, 1, 2, 3\}$. Following (Grill et al., 2020), we use an online and a target network to extract features from different views of the same sample, then a predictor maps the online projection to align with the target projection. Meanwhile, we attach a linear classification head $\omega \in \mathbb{R}^{d \times 4}$ to the online projection of $\tilde{x}$ to predict its rotation label, trained via the cross-entropy between the output logits and the true label $\varphi$. The representation alignment loss is therefore formulated as

$$\ell_{\text{rep}} = \frac{1}{2B} \sum_{i=1}^{B} \sum_{v=1}^{2} \left( \left\| q_\theta(\mathbf{z}_i^{(v)}) - \bar{\mathbf{z}}_i^{(v')} \right\|_2^2 + \text{CE}\big(\omega(\tilde{\mathbf{z}}_i^{(v)}), \varphi_i\big) \right), \tag{4}$$

where $v' = 3 - v$ denotes the other augmented view, $\bar{\mathbf{z}}$ denotes the projection outputted by target network with stop-gradient applied, and $\text{CE}(\cdot, \cdot)$ is the cross-entropy function. This combined objective encourages the online network to learn view-invariant and discriminative features, while the rotation prediction task provides an additional supervisory signal to mitigate collapse in the early stage.

### 4.3 SERVER-SIDE MODEL AGGREGATION

In each communication round, once participating clients complete local training, they upload both model parameters and local prototypes to the server. Model parameters are aggregated via weighted averaging, using each client's number of local samples as the aggregation weight. As local prototypes represent semantic centers specific to local data distributions, simply averaging would blur these distinct semantic clusters and destroy the learned structure. Therefore, we propose an optimization-based aggregation mechanism that treats the collection of local prototypes as training data to refine the global prototypes as learnable parameters, thereby consolidating local prototypes into an updated set of global prototypes.

Let $\mathbf{P}_l \in \mathbb{R}^{I \times d}$ denote the consolidated matrix of local prototypes collected from all participating clients, and $\mathbf{P}_g \in \mathbb{R}^{K \times d}$ denote the global prototypes. We formulate the aggregation of local prototypes as an optimization objective to align global prototypes with the consensus of local prototypes. Adopting a strategy analogous to the client-side semantic alignment, we treat the consolidated local prototypes $\mathbf{P}_l$ as the samples to be assigned to the global prototypes $\mathbf{P}_g$. We reuse the formulations to compute the similarity matrix $\mathbf{S}$ between $\mathbf{P}_l$ and $\mathbf{P}_g$, and the soft assignment matrix $\mathbf{Q}$ that indicates how strongly each local prototype is assigned to each global prototype, but with a key distinction in the constraints. Unlike the client-side UOT, which handles data heterogeneity, here we enforce strict equipartition constraints on both marginals via the balanced version of equation 2 to ensure that global prototypes uniformly cover the aggregated local semantic space. The primary objective, the assignment fidelity loss, is formulated as the cross-entropy between the assignment and similarity:

$$\ell_{\text{fed}} = -\frac{1}{I} \sum_{i=1}^{I} \sum_{k=1}^{K} \mathbf{Q}_{i,k} \log \mathbf{S}_{i,k}, \tag{5}$$

which encourages the similarity distribution to match the assignment. Minimizing $\ell_{\text{fed}}$ pulls each global prototype towards the centroid of local prototypes that are strongly assigned to it, ensuring the updated global prototypes reflect the consensus of the local semantic centers. To prevent the global prototypes from collapsing into a few redundant clusters and preserve the overall semantic diversity, we incorporate a prototype uniformity loss as a regularizer. This term penalizes excessive proximity via pairwise repulsion, i.e.,

$$\ell_{\text{uni}} \;=\; \log\left(\frac{1}{K(K-1)}\sum_{k \neq j}\exp\left(-2\gamma\,\|\mathbf{p}_k - \mathbf{p}_j\|^2\right)\right), \tag{6}$$

where $\gamma > 0$ controls the sharpness of the exponential weighting. The server employs an SGD optimizer to perform $U$ epochs of iterative updates on $\mathbf{P}_g$ by minimizing the combined objective $\mathcal{L}_{\text{proto}} = \ell_{\text{fed}} + \beta\,\ell_{\text{uni}}$, where $\beta$ is a trade-off coefficient. This optimization-based strategy ensures that the updated global prototypes are both representative of local semantics and globally well-separated. After aggregation, the updated global prototypes are distributed to clients for the next round of training, providing refined and globally consistent semantic guidance.

## 5 CONVERGENCE ANALYSIS

In this section, we provide a convergence analysis for our proposed federated unsupervised learning framework. We first detail the assumptions that underpin our analysis and then present our main theorems: one characterizing the sufficient decrease achieved by our prototype aggregation method and the other guaranteeing the global convergence rate of the entire algorithm. Our analysis quantitatively demonstrates how factors influence the final solution quality and precise bounds are provided in theorems below.

To facilitate our theoretical analysis, we introduce the following notation. Let $\theta$ denote the model parameters and $\rho$ denote the prototype parameters, and both are optimized jointly during local training. Thus, we define the combined parameter $\psi = (\theta, \rho)$. Any assumption stated for $\psi$ is understood to hold for both $\theta$ and $\rho$. Clients perform local updates with learning rate $\eta_l$, while the server performs $U$ steps of SGD with learning rate $\eta_\rho$ on the surrogate objective $F_s(\cdot)$. For brevity, detailed statements of the assumptions and the complete convergence proof are provided in Appendix B.

**Theorem 1** (Sufficient Decrease of Prototype Aggregation). *Let Assumption 1 and 6 hold, and the server-side learning rate satisfies $\eta_\rho \leq 1/L$, then the proposed prototype optimization step yields*

$$\mathbb{E}[F(\theta_r, \rho_{r+1})] \leq F(\theta_r, \rho_r) - \frac{\eta_\rho}{2}\sum_{u=0}^{U-1}\mathbb{E}\|\nabla_\rho F(\theta_r, \rho_u)\|^2 + \frac{\eta_\rho U \zeta^2}{2}.$$

This theorem formally bridges the gap between the server's surrogate optimization task and the true global objective. It guarantees that our prototype aggregation strategy achieves a sufficient decrease in the global loss each round, thus providing a principled convergence guarantee even when the server operates with limited information based on not exactly the same objective.

**Theorem 2** (Global Convergence). *Let Assumptions 1-6 hold and the server-side learning rate satisfies $\eta_\rho \leq 1/L$. After $R$ communication rounds, the average expected squared norm of the global gradient is bounded as follows:*

$$\frac{1}{R}\sum_{r=0}^{R-1}\mathbb{E}\|\nabla F(\psi_r)\|^2 \leq \frac{F(\psi_0) - F^\star}{R\,\Gamma_1} + \frac{\Gamma_2}{\Gamma_1},$$

*where $F^*$ is the minimum value of the global objective, $\Gamma_1 = \min(\frac{\eta_l E M}{2N}, \frac{\eta_\rho}{2})$ and $\Gamma_2 = \frac{G^2 M}{N}\left(\frac{L^2\eta_l^2 E^2}{2} + \frac{\eta_l^3 E^3 L^2}{3}\right) + \left(\eta_l + \frac{L^2\eta_l^2 EM}{2N} + \frac{\eta_l^3 E^2 L^2 M}{2N}\right)\bar{\sigma}^2 + \frac{\eta_\rho U \zeta^2}{2}.$*

This theorem demonstrates that our algorithm converges to a neighborhood of a stationary point at a sublinear rate of $\mathcal{O}(\frac{1}{R})$. Specifically, the size of this neighborhood is influenced by the state of the initial model, data heterogeneity across clients, the number of epochs, learning rates and the error from our prototype aggregation scheme, quantifying the trade-offs inherent in federated learning.

## 6 EXPERIMENTS

### 6.1 EXPERIMENTAL SETUP

**Datasets and partitioning.** We conduct experiments on CIFAR-10 and CIFAR-100 (Krizhevsky et al., 2009). To simulate non-IID data distributions across clients, we partition the training set among $N$ clients using a Dirichlet distribution (Hsu et al., 2019). Specifically, for each class we sample a probability vector from $\mathrm{Dir}(\alpha)$ and distribute samples to clients accordingly. The concentration parameter $\alpha$ controls the heterogeneity, with a smaller $\alpha$ resulting in more skewed partitions.

**Evaluation Protocol.** We evaluate representation quality using linear probing (Chen et al., 2020), K-nearest neighbors (KNN) classification (Chen & He, 2021) and semi-supervised fine-tuning. Linear probing trains a linear classifier on frozen features to assess the linear separability of the representations, while KNN classification on frozen embeddings provides a label-free measure of feature discriminability. Semi-supervised fine-tuning with 1% or 10% labeled data evaluates representation transferability in low-label regimes.

**Comparative Methods.** We compare **FedPAC** with several relevant baselines: (1) federated adaptations of centralized self-supervised learning methods, including SimCLR, BYOL, and SimSiam, combined with FedAvg, (2) the state-of-the-art federated unsupervised learning methods including FedU (Zhuang et al., 2021), FedX (Han et al., 2022), FedEMA (Zhuang et al., 2022), Orchestra (Lubana et al., 2022), FedU2 (Liao et al., 2024), and SSD (Fang et al., 2025). For fair comparison, all methods use the same encoder architecture and data partitions. Implementations follow the original papers and, where available, rely on official codebases. Results are averaged over 3 independent runs with different random seeds.

### 6.2 EXPERIMENT RESULTS

**Representation Evaluation.** Following existing methods (Liao et al., 2024; Lubana et al., 2022), we first assess the quality of the learned representations via linear probing and semi-supervised fine-tuning. The comprehensive results are presented in Table 1 and key observations are summarized as follows. **First**, simply combining centralized SSL methods with FedAvg yields limited accuracy under high data heterogeneity, confirming their fragility to non-IID data. **Second**, FedPAC consistently outperforms all baselines in linear evaluation across both cross-silo and cross-device settings, demonstrating the superior quality of learned representations. **Third**, under the challenging 1% semi-supervised setting, FedPAC achieves a substantial performance margin over compared methods. This can be attributed to its well-structured feature space, which already exhibits distinct and semantically coherent clusters. Consequently, downstream adaptation task is simplified to learning a linear mapping from these clusters to their corresponding labels, requiring minimal supervision. While this gap narrows as the proportion of labeled data increases to 10%, FedPAC remains highly competitive. The superior performance of FedPAC reported above underscores its ability to learn discriminative and transferable representations.

**Analysis of Sensitivity to hyperparameters.** We begin by evaluating the sensitivity of different methods to data heterogeneity, controlling $\alpha = \{0.1, 0.5, 1.0\}$. Results in Figure 2 show that FedPAC maintains stable performance even under severe non-IID settings, whereas the accuracy of baseline methods degrades, particularly on the more complex CIFAR-100 dataset. This demonstrates the robustness of our prototype-anchored consensus against skewed data distributions. For additional experiments analyzing the effects of other parameters, please refer to the Appendix C.

**Ablation Study** We conduct an ablation study under cross-device setting to evaluate the contribution of each component in FedPAC. As shown in Table 2, both the semantic alignment loss and the representation alignment loss are essential for achieving optimal performance. The semantic alignment loss contributes the most to cross-client consistency, while the representation alignment loss is critical in preventing representation collapse. The prototype aggregation strategy is also necessary for maintaining global semantic consensus. Removing any component leads to performance degradation, which validates our design choices. We also plot the convergence curves of FedPAC and FedU2 in Figure 3. Compared to FedU2, our method exhibits faster convergence and a more

Table 1: Accuracy (%) on CIFAR10 and CIFAR100 under non-IID ($\alpha = 0.1$) cross-device (100 clients) and cross-silo (10 clients) settings. FedU2 can be combined with different SSL methods and we list the results for all of them (denoted by superscripts). Evaluation is performed via linear probing and semi-supervised fine-tuning with 1%/10% labelled data.

| Dataset | CIFAR-10 | | | | | | CIFAR-100 | | | | | |
|---|---|---|---|---|---|---|---|---|---|---|---|---|
| Setting | Cross-Device (N=100) | | | Cross-Silo (N=10) | | | Cross-Device (N=100) | | | Cross-Silo (N=10) | | |
| Method | Linear | 1% | 10% | Linear | 1% | 10% | Linear | 1% | 10% | Linear | 1% | 10% |
| SimCLR | 61.70 | 48.76 | 68.82 | 74.63 | 63.34 | 76.25 | 34.42 | 13.25 | 36.61 | 50.42 | 19.25 | 43.61 |
| BYOL | 60.91 | 51.46 | 68.10 | 75.52 | 75.55 | 81.93 | 29.74 | 11.56 | 33.13 | 48.87 | 20.52 | 43.15 |
| SimSiam | 63.63 | 54.81 | 71.14 | 78.51 | 70.32 | 78.33 | 32.96 | 12.71 | 35.90 | 50.38 | 23.71 | 43.96 |
| FedU | 59.72 | 50.59 | 72.42 | 80.03 | 69.51 | 83.15 | 31.74 | 12.09 | 34.08 | 54.36 | 30.97 | 47.46 |
| FedX | 68.38 | 58.55 | 73.59 | 74.93 | 67.47 | 81.38 | 35.78 | 15.91 | 36.30 | 49.02 | 19.57 | 41.69 |
| FedEMA | 63.80 | 53.21 | 73.56 | 78.95 | 68.65 | 81.87 | 32.49 | 12.95 | 36.29 | 51.66 | 32.35 | 47.21 |
| Orchestra | 64.28 | 53.69 | 69.32 | 76.13 | 75.80 | **85.70** | 27.66 | 11.95 | 33.21 | 52.81 | 33.70 | 48.89 |
| FedU2$^{SimCLR}$ | 65.58 | 54.50 | 72.44 | 80.43 | 73.47 | 82.66 | 36.07 | 16.25 | 35.77 | 53.55 | 31.98 | 48.71 |
| FedU2$^{SimSiam}$ | 69.01 | 60.71 | 73.97 | 82.19 | 73.60 | 82.48 | 34.18 | 12.81 | 34.25 | 54.05 | 30.57 | 50.93 |
| FedU2$^{BYOL}$ | 68.81 | 56.69 | 72.56 | 82.80 | 75.04 | 84.67 | 36.25 | 17.01 | **38.00** | 54.78 | 32.19 | **52.69** |
| SSD | 67.18 | 58.60 | 74.43 | 80.85 | 74.85 | 83.28 | 36.28 | 16.38 | 36.35 | 54.62 | 35.71 | 50.15 |
| **FedPAC** | **71.36** | **62.17** | **75.03** | **83.56** | **77.59** | 84.47 | **37.32** | **17.68** | 37.64 | **56.33** | **37.23** | 51.98 |

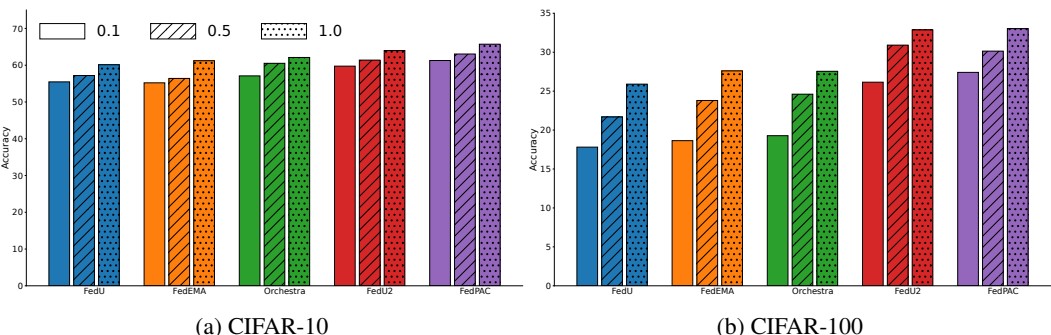

(a) CIFAR-10          (b) CIFAR-100

Figure 2: Sensitivity to data heterogeneity on CIFAR-10 (left) and CIFAR-100 (right). FedPAC is more robust to data heterogeneity.

stable training process with less oscillations throughout. In contrast, the variant of FedPAC without $\ell_{rep}$ shows slower accuracy improvement during early training and suffers from instability, further confirming its importance in improving representation quality and stabilizing training.

Table 2: Comparison of FedPAC and its ablated variants on CIFAR10 and CIFAR100 under non-IID ($\alpha = 0.1$) cross-device settings.

| Dataset | Method | FedPAC | w/o $\ell_{sem}$ | w/o $\ell_{rep}$ | w/o $\mathcal{L}_{proto}$ |
|---|---|---|---|---|---|
| CIFAR-10 | KNN | 63.99 | 55.91 | 59.87 | 61.48 |
| | Linear | 71.36 | 62.55 | 68.40 | 69.22 |
| | 1% | 62.17 | 53.36 | 60.40 | 60.62 |
| | 10% | 75.03 | 71.37 | 73.55 | 73.09 |
| CIFAR-100 | KNN | 27.41 | 19.68 | 20.16 | 23.02 |
| | Linear | 37.32 | 30.40 | 30.33 | 33.54 |
| | 1% | 17.68 | 12.99 | 12.01 | 14.01 |
| | 10% | 37.64 | 30.28 | 30.86 | 31.45 |

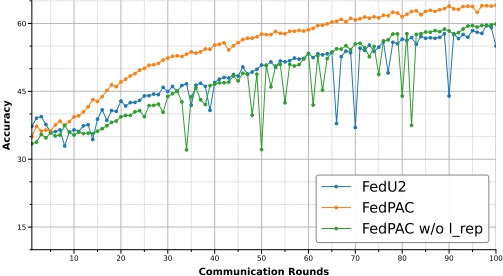

Figure 3: Convergence curve of KNN accuracy versus communication rounds on CIFAR-10.

## 7 CONCLUSION

In this work, we propose FedPAC to address the critical challenges of representation collapse and semantic misalignment in federated unsupervised learning under non-IID data. It leverages prototypes

as semantic anchors to establish a semantic consensus among clients, enables learning discriminative and semantically consistent representations from distributed, unlabeled, and imbalanced data. Theoretically we provide a rigorous convergence analysis, and empirically, we conduct experiments on CIFAR10 and CIFAR100 to validate the superior performance of FedPAC.

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

# A  METHOD DETAILS

## A.1  DETAILS ON PROTOTYPE ASSIGNMENT COMPUTATION

In this section, we provide a detailed derivation of the optimal transport formulations used in our framework. Given batch projections $\mathbf{Z} \in \mathbb{R}^{B \times d}$ and the prototypes $\mathbf{P} \in \mathbb{R}^{K \times d}$, we seek an assignment matrix $\mathbf{Q} \in \mathbb{R}^{B \times K}$ where $\mathbf{Q}_{i,k}$ denotes the mass that sample $i$ places on prototype $k$. We formulate this as an entropy-regularized unbalanced optimal transport problem. We start with the general formulation and then discuss two specific distinct variants tailored to the specific requirements of client-side training and server-side aggregation.

**General Formulation.**  The general entropy-regularized UOT problem relaxes both marginal constraints using divergence penalties:

$$\min_{\mathbf{Q} \in \mathbb{R}_+^{B \times K}} \langle \mathbf{Q}, \mathbf{C} \rangle - \varepsilon H(\mathbf{Q}) + \mu \operatorname{KL}(\mathbf{Q}^\top \mathbf{1}_B \,\|\, \mathbf{a}) + \nu \operatorname{KL}(\mathbf{Q}\mathbf{1}_K \,\|\, \mathbf{b}), \tag{7}$$

where cost matrix $\mathbf{C} = -\mathbf{Z}\mathbf{P}^\top$. Here $\mathbf{Q}\mathbf{1}_K \in \mathbb{R}^B$ and $\mathbf{Q}^\top \mathbf{1}_B \in \mathbb{R}^K$ are the sample and prototype side marginals respectively, while $\mathbf{a} \in \mathbb{R}^B$ and $\mathbf{b} \in \mathbb{R}^K$ are their target margins. The parameters $\mu, \nu \geq 0$ control the strength of the marginal constraints.

**Semi-relaxed UOT for local training.**  To address the heterogeneous setting, we preserve a hard marginal on the sample side to ensure that each sample is equally assigned. This is implemented by taking $\nu \to \infty$ and hence imposing $\mathbf{Q}\mathbf{1}_K = \mathbf{a} = \frac{1}{B}\mathbf{1}_B$. We relax the prototype-side marginal via a KL penalty with target $\mathbf{b} = \frac{1}{K}\mathbf{1}_K$. Thus, we can rewrite equation 7 as:

$$\min_{\mathbf{Q} \in \mathbb{R}_+^{B \times K}} \operatorname{Tr}\left( - \mathbf{Q}^\top \mathbf{Z}\mathbf{P}^\top \right) - \varepsilon H(\mathbf{Q}) + \mu \operatorname{KL}\left(\mathbf{Q}^\top \mathbf{1}_B \,\|\, \frac{1}{K}\mathbf{1}_K\right) \quad s.t. \ \mathbf{Q}\mathbf{1}_K = \frac{1}{B}\mathbf{1}_B. \tag{8}$$

Compared with strict equipartition, the one-sided unbalanced marginal constraint prevents trivial collapse while allowing prototype marginals to deviate from exact uniformity. This flexibility avoids spurious assignments when clients lack certain classes, produces more reliable assignments under non-IID data and still encourages balanced prototype utilization.

With entropy regularization and KL relaxation, $\mathbf{Q}^\star$, the solution of equation 8 admits a Gibbs-like factorization (Cuturi, 2013), i.e, $\mathbf{Q}^\star = \operatorname{diag}(\mathbf{u})\, \mathbf{G} \operatorname{diag}(\mathbf{v})$ with the Gibbs kernel matrix $\mathbf{G} = \exp(\mathbf{Z}\mathbf{P}^\top / \varepsilon)$. For numerical stability, we compute the multiplicative renormalizers $\mathbf{u}, \mathbf{v}$ by iterating the following calculations in the logarithmic domain:

$$\log \mathbf{u} \leftarrow \log \mathbf{a} - \log(\mathbf{G}\mathbf{v}), \qquad \log \mathbf{v} \leftarrow \kappa\left( \log \mathbf{b} - \log(\mathbf{G}^\top \mathbf{u})\right), \tag{9}$$

where $\kappa = \mu/(\mu + \varepsilon)$. After convergence we plug $\mathbf{u}$ and $\mathbf{v}$ back into the original factorization to obtain $\mathbf{Q}^\star$. Optionally one may obtain a discrete assignment from $\mathbf{Q}^\star$ by a rounding procedure, but we retain the continuous soft assignment in training because it provides smoother gradients and better numerical stability (Asano et al., 2020).

**Balanced OT for Global Prototype Aggregation.**  In contrast to local training, when aggregating prototypes on the server, we aim to maintain a globally diverse set of prototypes that uniformly covers the feature space. In this scenario, we enforce strict equipartition constraints on both samples and prototypes to maximize entropy and prevent collapse at the global level. This corresponds to the setting both that $\mu \to \infty$ and $\nu \to \infty$, which is equivalent to standard Optimal Transport with entropy regularization, i.e.,

$$\min_{\mathbf{Q} \in \mathbb{R}_+^{B \times K}} \operatorname{Tr}\left( - \mathbf{Q}^\top \mathbf{Z}\mathbf{P}^\top \right) - \varepsilon H(\mathbf{Q}) \quad s.t. \ \mathbf{Q}\mathbf{1}_K = \frac{1}{B}\mathbf{1}_B, \mathbf{Q}^\top \mathbf{1}_B = \frac{1}{K}\mathbf{1}_K. \tag{10}$$

The renormalizers in this case can be computed using the iterative Sinkhorn-Knopp algorithm (Cuturi, 2013).

---

**Algorithm 1** The FedPAC Framework

---

**Input:** Number of clients $N$, communication round $R$, local training epochs on clients $E$, prototypes optimization epochs on server $U$, local data $D_n$, trade-off coefficient $\lambda$, $\beta$.
**Output:** Global model $\theta_R$.
1: **Server executes:**
2: Initialize global model $\theta_0$ and global prototypes $\mathbf{P}_{g,0}$;
3: **for** each communication round $r = 0, 1, \ldots, R-1$ **do**
4:     Randomly select a subset of clients $S_r$;
5:     **for** each client $n \in S_r$ **in parallel do**
6:         Send $\theta_r$, $\mathbf{P}_{g,r}$ to client $n$;
7:         $\theta_n, \mathbf{P}_n \leftarrow$ **Client executes**($n$, $\theta_r$, $\mathbf{P}_{g,r}$);
8:     **end for**
9:     Aggregate models parameters: $\theta_{r+1} \leftarrow \text{FedAvg}(\theta_n|_{n \in S_r})$;
10:     Initialize $\mathbf{P}_{g,r+1} \leftarrow \mathbf{P}_{g,r}$ and collect local prototypes $\mathbf{P}_l \leftarrow \{\mathbf{P}_n\}_{n \in S_r}$;
11:     **for** prototypes optimization epoch $u = 1, 2, \ldots, U$ **do**
12:         Compute assignment fidelity loss $\ell_{\text{fed}}$ on $\mathbf{P}_{g,r+1}$ and $\mathbf{P}_l$ using equation 5;
13:         Compute prototype uniformity loss $\ell_{\text{uni}}$ on $\mathbf{P}_{g,r+1}$ using equation 6;
14:         Update global prototypes $\mathbf{P}_{g,r+1}$ by minimizing $\mathcal{L}_{\text{proto}} = \ell_{\text{fed}} + \beta\,\ell_{\text{uni}}$;
15:     **end for**
16: **end for**
17: **return** $\theta_R$
18: **Client executes**($n$, $\theta_r$, $\mathbf{P}_{g,r}$):
19: $\theta_n \leftarrow \theta_r$, $\mathbf{P}_n \leftarrow \mathbf{P}_{g,r}$;
20: **for** each local epoch $e = 1, 2, \ldots, E$ **do**
21:     Sample a mini-batch from $D_n$;
22:     Compute semantic alignment loss $\ell_{sem}$ using equation 3;
23:     Compute representation alignment $\ell_{rep}$ using equation 4;
24:     Update local model $\theta_n$ and prototypes $\mathbf{P}_n$ by minimizing $\mathcal{L}_{\text{local}} = \ell_{\text{sem}} + \lambda\,\ell_{\text{rep}}$;
25: **end for**
26: **return** updated $\theta_n$ and $\mathbf{P}_n$ back to the server

---

**Working on small batches.** When the batch size $B$ is much smaller than the number of prototypes $K$, an equal partitioning of the batch samples across $K$ prototypes is infeasible. To mitigate this issue, we augment the current batch with a memory queue containing the projections from recent training samples. We solve the UOT problem over this augmented set and only the entries of the assignment matrix corresponding to the samples from the current batch are utilized to compute the semantic alignment loss. This memory-augmented strategy enables the estimation of stable assignments with small batch sizes, while imposing minimal computational and memory overhead.

## A.2 NOTATION AND ALGORITHM

We present Table 3 to better summarize and explain the notations used in this paper. And we also summarize the entire framework in Algorithm 1 that better illustrates the entire training process.

## A.3 IMPLEMENTATION DETAILS

We adopt ResNet-18 (He et al., 2016) as the encoder architecture. The projector and predictor architectures, along with the EMA update rule, follow standard SSL protocols (Grill et al., 2020). Unless otherwise specified, we set the number of local epochs $E = 10$, total communication rounds $R = 100$, and the Dirichlet parameter $\alpha = 0.1$ to simulate high data heterogeneity. Our evaluation covers two FL scenarios: a cross-silo setting ($N = 10$ clients, 100% participation) and a cross-device setting ($N = 100$ clients, 10% participation). For optimization, we use SGD with a learning rate of 0.01 for both client-side training and server-side aggregation, where the latter is performed for $U = 200$ epochs. Regarding dataset-specific settings, we use a batch size of 64 with 32 prototypes for CIFAR-10, and a batch size of 128 with 128 prototypes for CIFAR-100. Table 4 details the specific values of other method-specific hyperparameters introduced in FedPAC.

Table 3: Summary of key notations used in this paper.

| Notation | Explanation |
|---|---|
| $R, r$ | Total number of communication rounds, current round |
| $N, n$ | Total number of clients, local client index |
| $\theta_r$ | Global model parameters at $r$-th round |
| $f_\theta,\ g_\theta,\ q_\theta$ | Online encoder, online projector and predictor parameterized by $\theta$ |
| $f_\phi,\ g_\phi$ | Target encoder and target projector parameterized by $\phi$ |
| $\mathbf{z}$ | Representation vector |
| $D_n$ | Local dataset of client $n$ |
| $x_{n,i}$ | The $i$-th sample of client $n$ (original input) |
| $\mathcal{T}$ | Stochastic augmentation function |
| $\tilde{x} = t(x),\ t \sim \mathcal{T}$ | Apply a random data transform to $x$ to get the augmented view |
| $E$ | The number of epochs for local training on each client |
| $B$ | Batch size for local training |
| $K$ | Total number of global prototypes |
| $\mathbf{P}, \mathbf{p}_k$ | Global prototype matrix, $k$-th prototype vector |
| $d$ | Dimensions of representation and prototype vectors |
| $\mathbf{Z}$ | The representation matrix of a batch |
| $\mathbf{S}$ | Similarity probability matrix |
| $\mathbf{Q}$ | Prototype assignments |
| $I$ | Total number of local prototypes from all participating clients in a round |
| $\rho$ | Parameterized representation of prototype vectors |
| $\psi = (\theta, \rho)$ | Joint parameterized representation of models and prototypes |
| $U$ | The number of epochs for optimizing global prototypes on the server |
| $F_s(\cdot)$ | Server-side surrogate function for optimizing global prototypes |
| $\eta_l, \eta_\rho$ | Learning rate for clients' local training, and server-side optimization |

Table 4: Detailed hyperparameter settings used in FedPAC experiments.

| Parameter | Value | Parameter | Value |
|---|---|---|---|
| *Client-side Training* | | *Server-side Aggregation* | |
| Entropic parameter $\epsilon$ | 0.05 | Entropic parameter $\epsilon$ | 0.1 |
| Trade-off coefficient $\lambda$ | 0.5 | Trade-off coefficient $\beta$ | 2.0 |
| KL penalty strength $\mu$ | 2.0 | Scale parameter $\gamma$ | 2.0 |
| Temperature parameter $\tau$ | 0.05 | Temperature parameter $\tau$ | 0.1 |
| Sinkhorn iterations | 5 | Sinkhorn iterations | 10 |

## A.4 DISCUSSION

**Comparison.** In Table 5, we systematically compare FedPAC with various state-of-the-art federated unsupervised learning approaches. As evident from the comparison, FedPAC distinguishes itself by establishing explicit semantic alignment across clients through shared prototypes. In contrast, prior methods typically rely on implicit alignment signals, such as distillation or parameter averaging, which often falter under severe heterogeneity. Furthermore, unlike clustering-based approaches like Orchestra that enforce hard constraints to compute assignments, which can lead to spurious assignments when clients lack specific classes, FedPAC handles data heterogeneity by relaxing marginal constraints through unbalanced optimal transport. Finally, FedPAC is underpinned by a rigorous theoretical foundation, offering a convergence guarantee that is rarely provided in existing FUL literature.

**Privacy.** While the primary objective of this work is learning representation in federated unsupervised learning, we recognize the concomitant privacy implications. FedPAC entails the transmission

Table 5: Comparison of properties between FedPAC and representative federated unsupervised learning methods.

| Method | Cross-silo | Cross-device | Heterogeneity Handling | Semantic Alignment | Convergence Guarantee |
|---|---|---|---|---|---|
| FedCA | ✓ | × | ✓ | × | × |
| FedU | ✓ | × | ✓ | × | × |
| FedEMA | ✓ | × | ✓ | × | × |
| Orchestra | ✓ | ✓ | ✓ | × | ✓ |
| FedX | ✓ | × | ✓ | × | × |
| FedU2 | ✓ | ✓ | ✓ | × | ✓ |
| SSD | ✓ | ✓ | ✓ | × | × |
| **FedPAC** | ✓ | ✓ | ✓ | ✓ | ✓ |

of both model parameters and local prototypes. While the exchange of model parameters is inherent to standard Federated Learning, prototypes serve as highly compressed statistical representations and the privacy risk associated with their transmission is significantly lower than sharing raw data. Moreover, the FedPAC framework is designed to be compatible with standard privacy-preserving mechanisms. Although these modules were not included in the current study to isolate the effects of our proposed method, they can be seamlessly integrated into the pipeline. For instance, Differential Privacy (DP) (Dwork et al., 2006; Abadi et al., 2016) can be applied by injecting Gaussian or Laplacian noise into parameters and prototypes prior to transmission. Similarly, Secure Aggregation (SecAgg) (Bonawitz et al., 2017) and Secure Multi-Party Computation (SMPC) (Yao, 1982)protocols can be employed to aggregate updates, ensuring the server accesses only the global aggregation without inspecting individual contributions. We regard the comprehensive investigation of the trade-off between representation learning and privacy constraints as a valuable direction for future research.

# B    PROOF OF THEOREMS

Let $\theta$ denote the model parameters and $\rho$ denote the prototype parameters, and both are optimized jointly during local training. Thus, we define the combined parameter $\psi = (\theta, \rho)$. Any assumption stated for $\psi$ is understood to hold for both $\theta$ and $\rho$. Clients perform local updates with learning rate $\eta_l$, while the server performs $U$ steps of SGD with learning rate $\eta_\rho$ on the surrogate objective $H(\cdot)$.

## B.1    ASSUMPTIONS

We base our convergence analysis on the following standard assumptions.

**Assumption 1** (Smoothness). *Local objective functions $F_1$, $F_2$, ... ,$F_N$ are all L-smooth, i.e.,* $\|\nabla F_n(\psi) - \nabla F_n(\psi')\| \le L\|\psi - \psi'\|$ *for $n = 1, \ldots, N$.*

**Assumption 2** (Unbiased Gradient). *Let $\xi$ denote a batch of samples uniformly sampled at random from local data. The stochastic gradient is an unbiased estimator of the true local gradient, i.e.,* $\mathbb{E}\left[\nabla F_n(\psi, \xi)\right] = \nabla F_n(\psi)$.

**Assumption 3** (Bounded Variance). *The variance of the stochastic gradient on each client $n$ is bounded:* $\mathbb{E}\|\nabla F_n(\psi, \xi) - \nabla F_n(\psi)\|^2 \le \sigma_n^2$ *for $n = 1, \ldots, N$.*

**Assumption 4** (Bounded Gradient Norm). *The expected squared norm of any client's stochastic gradient is uniformly bounded:* $\mathbb{E}\|\nabla F_n(\psi, \xi)\|^2 \le G^2$, *for $n = 1, \ldots, N$.*

**Assumption 5** (Uniform Client Sampling). *In each communication round $r$, a set $S_r$ of $M$ clients is selected uniformly at random from the total $N$ clients. The probability of any client $n$ being selected is* $\mathbb{P}(n \in S_r) = M/N$.

**Assumption 6** (Bounded Prototype Gradient Estimation Error). *The gradient of the server-side surrogate function $\nabla F_s(\rho_r)$ is an estimator of the true global gradient with bounded variance. Specifically, at communication round $r$, we have* $\mathbb{E}\|\nabla F_s(\rho_r) - \nabla_\rho F(\theta_r, \rho_r)\|^2 \le \zeta^2$.

The mean squared error between the server-side prototype gradient $\nabla H(\rho_r)$ and the true global prototype gradient $\nabla_\rho F(\theta_r, \rho_r)$ is bounded: $\mathbb{E}\left[\nabla H(\rho_r)\right] = \nabla_\rho F(\theta_r, \rho_r)$

## B.2 CONVERGENCE ANALYSIS

Let $\rho_0 = \rho_r$ and $\rho_U = \rho_{r+1}$ denote the initial and final prototype parameters at round $r$, respectively. The update rule for each step $u$ is:

$$\rho_{u+1} = \rho_u - \eta_\rho \nabla F_s(\rho_u). \tag{11}$$

We first prove Theorem 1, which establishes that our prototype aggregation strategy guarantees a sufficient decrease in the global objective function.

**Theorem 1** (Sufficient Decrease Guarantee of Prototype Aggregation)*. Let Assumption 1 and 6 hold, and the server-side learning rate satisfies $\eta_\rho \leq 1/L$, then the proposed prototype optimization step yields*

$$\mathbb{E}[F(\theta_r, \rho_{r+1})] \leq F(\theta_r, \rho_r) - \frac{\eta_\rho}{2} \sum_{u=0}^{U-1} \mathbb{E}\|\nabla_\rho F(\theta_r, \rho_u)\|^2 + \frac{\eta_\rho U \zeta^2}{2}.$$

*Proof.* With Assumption 1 holds, considering $\psi_r$ as fixed during the server-side prototype aggregation, it follows that

$$F(\theta_r, \rho_u) \leq F(\theta_r, \rho_u) + \langle \nabla_\rho F(\theta_r, \rho_u), \rho_{u+1} - \rho_u \rangle + \frac{L}{2}\|\rho_{u+1} - \rho_u\|^2. \tag{12}$$

Then substituting the update difference $\rho_{u+1} - \rho_u = -\eta_\rho \nabla F_s(\rho_u)$ into equation 12, we have

$$F(\theta_r, \rho_{u+1}) \leq F(\theta_r, \rho_u) - \eta_\rho \langle \nabla_\rho F(\theta_r, \rho_u), \nabla F_s(\rho_u) \rangle + \frac{L\eta_\rho^2}{2}\|\nabla F_s(\rho_u)\|^2. \tag{13}$$

Taking the expectation on both sides of the above formula, we have

$$\mathbb{E}[F(\theta_r, \rho_{u+1})] \leq F(\theta_r, \rho_u) - \eta_\rho \underbrace{\mathbb{E}\langle \nabla_\rho F(\theta_r, \rho_u), \nabla F_s(\rho_u) \rangle}_{A_1} + \frac{L\eta_\rho^2}{2}\mathbb{E}\|\nabla F_s(\rho_u)\|^2. \tag{14}$$

Using the identity $2\langle a, b \rangle = \|a\|^2 + \|b\|^2 - \|a - b\|^2$ and let $a = \nabla_\rho F(\theta_r, \rho_u)$ and $b = \nabla F_s(\rho_u)$, it follows that

$$A_1 = \mathbb{E}\left[\frac{1}{2}\left(\|\nabla_\rho F(\psi_r, \rho_u)\|^2 + \|\nabla F_s(\rho_u)\|^2 - \|\nabla_\rho F(\theta_r, \rho_u) - \nabla F_s(\rho_u)\|^2\right)\right]$$

$$= \frac{1}{2}\left(\mathbb{E}\|\nabla_\rho F(\theta_r, \rho_u)\|^2 + \mathbb{E}\|\nabla F_s(\rho_u)\|^2 - \mathbb{E}\|\nabla_\rho F(\theta_r, \rho_u) - \nabla F_s(\rho_u)\|^2\right).$$

Plugging back into equation 14, we have

$$\mathbb{E}[F(\theta_r, \rho_u)] \leq F(\theta_r, \rho_r) + \frac{L\eta_\rho^2}{2}\mathbb{E}\|\nabla F_s(\rho_u)\|^2$$

$$- \frac{\eta_\rho}{2}\left(\mathbb{E}\|\nabla_\rho F(\theta_r, \rho_u)\|^2 + \mathbb{E}\|\nabla F_s(\rho_u)\|^2 - \mathbb{E}\|\nabla_\rho F(\theta_r, \rho_u) - \nabla F_s(\rho_u)\|^2\right)$$

$$= F(\theta_r, \rho_r) - \frac{\eta_\rho}{2}\mathbb{E}\|\nabla_\rho F(\theta_r, \rho_u)\|^2 - \left(\frac{\eta_\rho}{2} - \frac{L\eta_\rho^2}{2}\right)\mathbb{E}\|\nabla F_s(\rho_u)\|^2$$

$$+ \frac{\eta_\rho}{2}\mathbb{E}\|\nabla_\rho F(\theta_r, \rho_u) - \nabla F_s(\rho_u)\|^2. \tag{15}$$

We choose the server-side learning rate such that $\eta_\rho \leq 1/L$, which implies that the coefficient of the $\mathbb{E}\|\nabla F_s(\rho_u)\|^2$ term is non-negative. We can thus drop this term to obtain a valid upper bound:

$$\mathbb{E}[F(\theta_r, \rho_{u+1})] \leq F(\theta_r, \rho_u) - \frac{\eta_\rho}{2}\mathbb{E}\|\nabla_\rho F(\theta_r, \rho_u)\|^2 + \frac{\eta_\rho}{2}\mathbb{E}\|\nabla_\rho F(\theta_r, \rho_u) - \nabla F_s(\rho_u)\|^2. \tag{16}$$

Applying Assumption 6 to bound the last term, we have

$$\mathbb{E}[F(\theta_r, \rho_{u+1})] \leq F(\theta_r, \rho_u) - \frac{\eta_\rho}{2}\mathbb{E}\|\nabla_\rho F(\theta_r, \rho_u)\|^2 + \frac{\eta_\rho \zeta^2}{2}. \tag{17}$$

Note that $\sum_{u=0}^{U-1}\left(\mathbb{E}\left[F(\rho_{u+1}) - F(\rho_r)\right]\right) = \mathbb{E}[F(\rho_{u+1})] - F(\rho_r)$, and summing over $u = 0, \ldots, U - 1$, we have

$$\mathbb{E}[F(\theta_r, \rho_{r+1})] \leq F(\theta_r, \rho_r) - \frac{\eta_\rho}{2} \sum_{u=0}^{U-1} \mathbb{E}\|\nabla_\rho F(\theta_r, \rho_u)\|^2 + \frac{\eta_\rho U \zeta^2}{2}. \tag{18}$$

This completes the proof.

$\square$

Before presenting the Theorem 2, we state and prove several lemmas to clearly express our subsequent proof clearly.

**Lemma 1** (Local Model Divergence). *Let Assumptions 2 to 4 hold, the expected squared deviation between the initial global model $\psi_r$ and local model parameters $\psi_n^E$ after $E$ local update steps is bounded as follows:*

$$\mathbb{E}\left[\|\psi_n^E - \psi_r\|^2\right] \leq \eta_l^2 E^2 G^2 + \eta_l^2 E \sigma_n^2$$

*Proof.* Note that the total change in the local model parameters on client $n$ after $E$ steps is the sum of the individual updates with local learning rate $\eta_l$, we have

$$\psi_n^E - \psi_r = \sum_{e=0}^{E-1} (\psi_n^{e+1} - \psi_n^e) = -\eta_l \sum_{e=0}^{E-1} \nabla F_n(\psi_n^e, \xi_n^e). \tag{19}$$

Taking the expected squared norm, we have

$$\mathbb{E}\|\psi_n^E - \psi_r\|^2 = \eta_l^2 \mathbb{E}\left\|\sum_{e=0}^{E-1} \nabla F_n(\psi_n^e, \xi_n^e)\right\|^2. \tag{20}$$

To analyze the sum of gradients, we decompose each stochastic gradient into the true local gradient and a zero-mean noise term $\delta_n^e$ such that

$$\nabla F_n(\psi_n^e, \xi_n^e) = \nabla F_n(\psi_n^e) + \delta_n^e, \tag{21}$$

where $\delta_n^e = \nabla F_n(\psi_n^e, \xi_n^e) - \nabla F_n(\psi_n^e)$. Plugging it into equation 20, we have

$$\mathbb{E}\|\psi_n^E - \psi_r\|^2 = \eta_l^2 \mathbb{E}\left\|\sum_{e=0}^{E-1} (\nabla F_n(\psi_n^e) + \delta_n^e)\right\|^2. \tag{22}$$

Then we expand the squared norm and get

$$\left\|\sum_{e=0}^{E-1} (\nabla F_n(\psi_n^e) + \delta_n^e)\right\|^2 = \left\|\sum_{e=0}^{E-1} \nabla F_n(\psi_n^e)\right\|^2 + \left\|\sum_{e=0}^{E-1} \delta_n^e\right\|^2 + 2\left\langle \sum_{e=0}^{E-1} \nabla F_n(\psi_n^e), \sum_{e=0}^{E-1} \delta_n^e \right\rangle. \tag{23}$$

Note that the expectation of the cross-term is zero with Assumption 2 hold. Thus, we have

$$\mathbb{E}\|\psi_n^E - \psi_r\|^2 = \eta_l^2 \left( \mathbb{E}\left\|\sum_{e=0}^{E-1} \nabla F_n(\psi_n^e)\right\|^2 + \mathbb{E}\left\|\sum_{e=0}^{E-1} \delta_n^e\right\|^2 \right). \tag{24}$$

Since the noise terms $\delta_n^e$ are independent across steps and have zero mean conditioned on the history, their cross terms vanish in expectation. Therefore, we can rewrite the second term as

$$\mathbb{E}\left\|\sum_{e=0}^{E-1} \delta_n^e\right\|^2 = \sum_{e=0}^{E-1} \mathbb{E}\|\delta_n^e\|^2$$

$$\leq \sum_{e=0}^{E-1} \sigma_n^2$$

$$= E\sigma_n^2, \tag{25}$$

where we use Assumption 3 to obtain the upper bound in the second step. By Cauchy-Schwarz inequality, for the first term we have

$$\mathbb{E}\left\|\sum_{e=0}^{E-1}\nabla F_n(\psi_n^e)\right\|^2 \leq E\sum_{e=0}^{E-1}\mathbb{E}\|\nabla F_n(\psi_n^e)\|^2. \tag{26}$$

Under Assumption 4, we know that $\mathbb{E}\|\nabla F_n(\psi_n^e, \xi_n^e)\|^2 \leq G^2$. Since $\mathbb{E}\|\nabla F_n(\psi_n^e, \xi_n^e)\|^2 = \mathbb{E}\|\nabla F_n(\psi_n^e) + \delta_n^e\|^2 = \mathbb{E}\|\nabla F_n(\psi_n^e)\|^2 + \mathbb{E}\|\delta_n^e\|^2$ and $\mathbb{E}\|\delta_n^e\|^2 \geq 0$, it follows that $\mathbb{E}[\|\nabla F_n(\psi_n^e)\|^2] \leq G^2$ and

$$\mathbb{E}\left\|\sum_{e=0}^{E-1}\nabla F_n(\psi_n^e)\right\|^2 \leq E\sum_{e=0}^{E-1}G^2 = E^2G^2. \tag{27}$$

Combining these two bounds with equation 24, we have

$$\mathbb{E}\|\psi_n^E - \psi_r\|^2 \leq \eta_l^2(E^2G^2 + E\sigma_n^2)$$
$$= \eta_l^2E^2G^2 + \eta_l^2E\sigma_n^2. \tag{28}$$

This completes the proof. $\qquad\square$

**Lemma 2** (Deviation of Local Stochastic Gradients). *Let Assumptions 1, 3 and 4 hold, the deviation between the average local stochastic gradient over $E$ steps and the true gradient at the initial model $\psi_r$ is bounded in expectation, i.e.,*

$$\mathbb{E}\left\|\frac{1}{E}\sum_{e=0}^{E-1}\nabla F_n(\psi_n^e, \xi_n^e) - \nabla F_n(\psi_r)\right\|^2 \leq \frac{2\sigma_n^2}{E} + \frac{2}{3}L^2\eta_l^2G^2E^2 + L^2\eta_l^2\sigma_n^2E$$

*Proof.* We can decompose the total discrepancy into two terms, $C_1$ and $C_2$:

$$\frac{1}{E}\sum_{e=0}^{E-1}\nabla F_n(\psi_n^e, \xi_n^e) - \nabla F_n(\psi_r) = \underbrace{\frac{1}{E}\sum_{e=0}^{E-1}(\nabla F_n(\psi_n^e, \xi_n^e) - \nabla F_n(\psi_n^e))}_{C_1}$$

$$+ \underbrace{\frac{1}{E}\sum_{e=0}^{E-1}(\nabla F_n(\psi_n^e) - \nabla F_n(\psi_r))}_{C_2}. \tag{29}$$

Using the inequality $\|a + b\|^2 \leq 2\|a\|^2 + 2\|b\|^2$, we can bound the expected squared norm as

$$\mathbb{E}\|C_1 + C_2\|^2 \leq 2\mathbb{E}\|C_1\|^2 + 2\mathbb{E}\|C_2\|^2. \tag{30}$$

We can rewrite the first term as

$$\mathbb{E}\|C_1\|^2 = \frac{1}{E^2}\mathbb{E}\left\|\sum_{e=0}^{E-1}(\nabla F_n(\psi_n^e, \xi_n^e) - \nabla F_n(\psi_n^e))\right\|^2 = \frac{1}{E^2}\mathbb{E}\left\|\sum_{e=0}^{E-1}\delta_n^e\right\|^2. \tag{31}$$

Using equation 25 again, and it follows that

$$\mathbb{E}\|C_1\|^2 \leq \frac{E\sigma_n^2}{E^2} = \frac{\sigma_n^2}{E}. \tag{32}$$

We can rewrite the second term as

$$\mathbb{E}\|C_2\|^2 = \mathbb{E}\left\|\frac{1}{E}\sum_{e=0}^{E-1}(\nabla F_n(\psi_n^e) - \nabla F_n(\psi_r))\right\|^2. \tag{33}$$

By Jensen's inequality, we have

$$\mathbb{E}\|C_2\|^2 \le \frac{1}{E} \sum_{e=0}^{E-1} \mathbb{E}\|\nabla F_n(\psi_n^e) - \nabla F_n(\psi_r)\|^2$$

$$\le \frac{L^2}{E} \sum_{e=0}^{E-1} \mathbb{E}\|\psi_n^e - \psi_r\|^2, \tag{34}$$

where we apply Assumption 1 to obtain the second inequality.

To bound the term $\mathbb{E}\left[\|\psi_n^e - \psi_r\|^2\right]$ for $e < E$, we use a result similar to Lemma 1, but for $e$ steps instead of $E$, i.e.,

$$\mathbb{E}\|\psi_n^e - \psi_r\|^2 \le \eta_l^2 e^2 G^2 + \eta_l^2 e \sigma_n^2. \tag{35}$$

Plugging this into Eq. equation 34, we have

$$\mathbb{E}\|C_2\|^2 \le \frac{L^2}{E} \sum_{e=0}^{E-1} \left(\eta_l^2 e^2 G^2 + \eta_l^2 e \sigma_n^2\right) \tag{36}$$

$$= \frac{L^2 \eta_l^2}{E} \left(G^2 \sum_{e=0}^{E-1} e^2 + \sigma_n^2 \sum_{e=0}^{E-1} e\right) \tag{37}$$

Using the formulas for the sum of integers and sum of squares, i.e., $\sum_{e=0}^{E-1} i = \frac{(E-1)E}{2}$ and $\sum_{e=0}^{E-1} i^2 = \frac{(E-1)E(2E-1)}{6}$, we have

$$\mathbb{E}\|C_2\|^2 \le \frac{L^2 \eta_l^2}{E} \left(G^2 \frac{(E-1)E(2E-1)}{6} + \sigma_n^2 \frac{(E-1)E}{2}\right)$$

$$\le \frac{L^2 \eta_l^2}{E} \left(G^2 \frac{E^3}{3} + \sigma_n^2 \frac{E^2}{2}\right)$$

$$= \frac{1}{3} L^2 \eta_l^2 G^2 E^2 + \frac{1}{2} L^2 \eta_l^2 \sigma_n^2 E. \tag{38}$$

Then we can combing equation 32 and equation 38 with equation 30 to obtain the bound for initial decomposition:

$$\mathbb{E}\left\|\frac{1}{E} \sum_{e=0}^{E-1} \nabla F_n(\psi_n^e, \xi_n^e) - \nabla F_n(\psi_r)\right\|^2 \le 2\left(\frac{\sigma_n^2}{E}\right) + 2\left(\frac{1}{3} L^2 \eta_l^2 G^2 E^2 + \frac{1}{2} L^2 \eta_l^2 \sigma_n^2 E\right)$$

$$= \frac{2\sigma_n^2}{E} + \frac{2}{3} L^2 \eta_l^2 G^2 E^2 + L^2 \eta_l^2 \sigma_n^2 E. \tag{39}$$

This completes the proof. $\qquad\square$

**Lemma 3** (Expectation of Random Client Sampling). *Let Assumption 5 hold and $X_n$ be a client-specific random quantity independent of the client selection process, the expectation of the weighted sum over the randomly selected client set $S_r$ satisfies that*

$$\mathbb{E}_{S_r}\left[\sum_{n \in S_r} w_n X_n\right] = \frac{M}{N} \sum_{n=1}^{N} w_n \mathbb{E}\left[X_n\right]$$

*Proof.* Let $I_n$ be a binary random variable indicating the participation of client $n$ in round $r$, where $I_n = 1$ if selected and 0 otherwise. With Assumption 5 holds, each client is selected independently with probability $M/N$ and it follows that $\mathbb{E}[I_n] = M/N$. The weighted sum over the randomly selected set $S_r$ can be rewritten using these indicators such that

$$\sum_{n \in S_r} w_n X_n = \sum_{n=1}^{N} I_n w_n X_n. \tag{40}$$

Take the expectation of the above formula and applying linearity of expectation, we have

$$\mathbb{E}_{S_r}\left[\sum_{n\in S_r} w_n X_n\right] = \mathbb{E}\left[\sum_{n=1}^{N} I_n w_n X_n\right] = \sum_{n=1}^{N}\mathbb{E}\left[I_n w_n X_n\right]. \tag{41}$$

If the client selection $I_n$ is independent of the client-specific quantity $X_n$, we have

$$\sum_{n=1}^{N}\mathbb{E}\left[I_n w_n X_n\right] = \sum_{n=1}^{N}\mathbb{E}\left[I_n\right]\cdot\mathbb{E}\left[w_n X_n\right] = \frac{M}{N}\sum_{n=1}^{N} w_n\mathbb{E}\left[X_n\right. \tag{42}$$

This completes the proof.

$$\square$$

**Theorem 2** (Global Convergence). *Let Assumptions 1-6 hold and the server-side learning rate satisfies $\eta_\rho \leq 1/L$. After $R$ communication rounds, the average expected squared norm of the global gradient is bounded as follows:*

$$\frac{1}{R}\sum_{r=0}^{R-1}\mathbb{E}\|\nabla F(\psi_r)\|^2 \leq \frac{F(\theta_0) - F^*}{R\Gamma_1} + \frac{\Gamma_2}{\Gamma_1},$$

*where $F^*$ is the minimum value of the global objective, $\Gamma_1$ and $\Gamma_2$ are constants that depend on problems' parameters and the algorithm's hyperparameters, but are independent of the total number of communication rounds $R$.*

*Proof.* With Assumption 1 holds, we have

$$\mathbb{E}[F(\psi_{r+1})] \leq \mathbb{E}[F(\psi_r)] + \mathbb{E}\langle\nabla F(\psi_r), \psi_{r+1} - \psi_r\rangle + \frac{L}{2}\mathbb{E}\|\psi_{r+1} - \psi_r\|^2. \tag{43}$$

Due to our hybrid aggregation strategy, we decompose the update $\psi_{r+1} - \psi_r = (\theta_{r+1} - \theta_r, \rho_{r+1} - \rho_r)$ and the gradient $\nabla F(\psi_r) = (\nabla_\theta F(\psi_r), \nabla_\rho F(\psi_r))$, it follows that

$$\mathbb{E}[F(\psi_{r+1})] \leq \mathbb{E}[F(\psi_r)] + \mathbb{E}\langle\nabla_\theta F(\psi_r), \theta_{r+1} - \theta_r\rangle + \mathbb{E}\langle\nabla_\rho F(\psi_r), \rho_{r+1} - \rho_r\rangle$$

$$+ \frac{L}{2}\mathbb{E}\|\theta_{r+1} - \theta_r\|^2 + \frac{L}{2}\mathbb{E}\|\rho_{r+1} - \rho_r\|^2$$

$$= \mathbb{E}[F(\psi_r)] + \underbrace{(\mathbb{E}[F(\theta_r, \rho_{r+1})] - F(\theta_r, \rho_r))}_{D_1}$$

$$+ \underbrace{\mathbb{E}\langle\nabla_\rho F(\psi_r), \rho_{r+1} - \rho_r\rangle + \frac{L}{2}\mathbb{E}\|\rho_{r+1} - \rho_r\|^2 - (\mathbb{E}[F(\theta_r, \rho_{r+1})] - F(\theta_r, \rho_r))}_{D_2}$$

$$+ \underbrace{\frac{L}{2}\mathbb{E}\|\theta_{r+1} - \theta_r\|^2}_{D_3} + \underbrace{\mathbb{E}\langle\nabla_\theta F(\psi_r), \theta_{r+1} - \theta_r\rangle}_{D_4}. \tag{44}$$

By Theorem 1, we know that

$$D_1 \leq -\frac{\eta_\rho}{2}\sum_{u=0}^{U-1}\mathbb{E}\|\nabla_\rho F(\theta_r, \rho_u)\|^2 + \frac{\eta_\rho U\zeta^2}{2}. \tag{45}$$

Under Assumption 1 and assuming $\theta_r$ fixed, we have

$$F(\theta_r, \rho_{r+1}) \leq F(\theta_r, \rho_r) + \langle\nabla_\rho F(\theta_r, \rho_r), \rho_{r+1} - \rho_r\rangle + \frac{L}{2}\|\rho_{r+1} - \rho_r\|^2. \tag{46}$$

Take the expectation of both sides and transpose the terms, we have

$$\mathbb{E}\langle\nabla_\rho F(\theta_r, \rho_r), \rho_{r+1} - \rho_r\rangle + \frac{L}{2}\mathbb{E}\|\rho_{r+1} - \rho_r\|^2 - \mathbb{E}[F(\theta_r, \rho_{r+1})] - F(\theta_r, \rho_r) \geq 0, \tag{47}$$

which is equivalent to $D_2 \geq 0$ and we can thus remove it from equation 44 to obtain the upper bound.

The update in $D_3$ is $\theta_{r+1} - \theta_r = \sum_{n \in S_r} w_n(\psi_n^E - \psi_r)$. Using Jensen's inequality, Lemma 1 and 3, we have

$$
\begin{aligned}
D_3 = \frac{L}{2} \mathbb{E} \left\| \sum_{n \in S_r} w_n(\psi_n^E - \psi_r) \right\|^2 &\leq \frac{LM}{2N} \sum_{n=1}^N w_n \mathbb{E} \|\psi_n^E - \psi_r\|^2 \\
&\leq \frac{LM}{2N} \sum_{n=1}^N w_n (\eta_l^2 G^2 E^2 + \eta_l^2 \sigma_n^2 E) \\
&= \frac{L^2 \eta_l^2 M}{2N} (G^2 E^2 + \bar{\sigma}^2 E),
\end{aligned}
\tag{48}
$$

where $\bar{\sigma}^2 = \sum_{n=1}^N w_n \sigma_n^2$.

And we have $\theta_{r+1} - \theta_r = \sum_{n \in S_r} w_n(\psi_n^E - \psi_r) = -\eta_l \sum_{n \in S_r} w_n \sum_{e=0}^{E-1} \nabla_\theta F_n(\theta_n^e, \xi_n^e)$, using Assumption 3, we have

$$
\begin{aligned}
D_4 &= -\eta_l E \cdot \mathbb{E} \left\langle \nabla_\theta F(\psi_r), \sum_{n \in S_r} w_n \left( \frac{1}{E} \sum_{e=0}^{E-1} \nabla_\theta F_n(\psi_n^e, \xi_n^e) \right) \right\rangle \\
&= -\frac{\eta_l M}{N} E \cdot \left\langle \nabla_\theta F(\psi_r), \sum_{n=1}^N w_n \mathbb{E} \left[ \frac{1}{E} \sum_{e=0}^{E-1} \nabla_\theta F_n(\psi_n^e, \xi_n^e) \right] \right\rangle
\end{aligned}
\tag{49}
$$

Using the identity $-\langle a, b \rangle \leq \frac{1}{2} \|a - b\|^2 - \frac{1}{2} \|a\|^2$, we have

$$
D_4 \leq -\frac{\eta_l E M}{2N} \|\nabla_\theta F(\psi_r)\|^2 + \frac{\eta_l E M}{2N} \mathbb{E} \left\| \nabla_\theta F(\psi_r) - \sum_{n=1}^N w_n \left( \frac{1}{E} \sum_{e=0}^{E-1} \nabla_\theta F_n(\psi_n^e, \xi_n^e) \right) \right\|^2.
\tag{50}
$$

Using Jensen's inequality and Lemma 2, we have

$$
\begin{aligned}
D_4 &\leq -\frac{\eta_l E M}{2N} \|\nabla_\theta F(\psi_r)\|^2 + \frac{\eta_l E M}{2N} \sum_{n=1}^N w_n \mathbb{E} \left\| \nabla_\theta F_n(\psi_r) - \left( \frac{1}{E} \sum_{e=0}^{E-1} \nabla_\theta F_n(\psi_n^e, \xi_n^e) \right) \right\|^2 \\
&\leq -\frac{\eta_l E M}{2N} \|\nabla_\theta F(\psi_r)\|^2 + \frac{\eta_l E M}{2N} \sum_{n=1}^N w_n \left( \frac{2\sigma_n^2}{E} + \frac{2}{3} L^2 \eta_l^2 G^2 E^2 + L^2 \eta_l^2 \sigma_n^2 E \right) \\
&= \frac{M}{N} \left( -\frac{\eta_l E}{2} \|\nabla_\theta F(\psi_r)\|^2 + \eta_l \bar{\sigma}^2 + \frac{1}{3} \eta_l^3 E^3 L^2 G^2 + \frac{1}{2} \eta_l^3 E^2 L^2 \bar{\sigma}^2 \right).
\end{aligned}
\tag{51}
$$

Combining equation 45, equation 48 and equation 51 with equation 44, we have

$$
\begin{aligned}
\mathbb{E}[F(\psi_{r+1})] \leq {} & \mathbb{E}[F(\psi_r)] - \frac{\eta_l E M}{2N} \|\nabla_\theta F(\psi_r)\|^2 - \frac{\eta_\rho}{2} \sum_{u=0}^{U-1} \mathbb{E} \|\nabla_\rho F(\theta_r, \rho_u)\|^2 \\
& + \frac{G^2 M}{N} \left( \frac{L^2 \eta_l^2 E^2}{2} + \frac{\eta_l^3 E^3 L^2}{3} \right) + \left( \eta_l + \frac{L^2 \eta_l^2 E M}{2N} + \frac{\eta_l^3 E^2 L^2 M}{2N} \right) \bar{\sigma}^2 \\
& + \frac{\eta_\rho U \zeta^2}{2}.
\end{aligned}
\tag{52}
$$

Note that

$$
\sum_{u=0}^{U-1} \mathbb{E} \|\nabla_\rho F(\theta_r, \rho_u)\|^2 \geq \mathbb{E} \|\nabla_\rho F(\theta_r, \rho_0)\|^2 = \|\nabla_\rho F(\theta_r, \rho_r)\|^2 = \|\nabla_\rho F(\psi_r)\|^2,
\tag{53}
$$

we can rewrite equation 52 as

$$\mathbb{E}[F(\psi_{r+1})] \leq \mathbb{E}[F(\psi_r)] - \frac{\eta_l EM}{2N}\|\nabla_\theta F(\psi_r)\|^2 - \frac{\eta_\rho}{2}\|\nabla_\rho F(\psi_r)\|^2 +$$
$$+ \frac{G^2 M}{N}\left(\frac{L^2\eta_l^2 E^2}{2} + \frac{\eta_l^3 E^3 L^2}{3}\right) + \left(\eta_l + \frac{L^2\eta_l^2 EM}{2N} + \frac{\eta_l^3 E^2 L^2 M}{2N}\right)\bar{\sigma}^2$$
$$+ \frac{\eta_\rho U\zeta^2}{2}. \tag{54}$$

Let $\Gamma_1 = \min(\frac{\eta_l EM}{2N}, \frac{\eta_\rho}{2})$ and $\Gamma_2$ collects all constant terms, i.e., $\Gamma_2 = \frac{G^2 M}{N}\left(\frac{L^2\eta_l^2 E^2}{2} + \frac{\eta_l^3 E^3 L^2}{3}\right) +$ $\left(\eta_l + \frac{L^2\eta_l^2 EM}{2N} + \frac{\eta_l^3 E^2 L^2 M}{2N}\right)\bar{\sigma}^2 + \frac{\eta_\rho U\zeta^2}{2}$, we can rearrange the inequality as

$$\Gamma_1 \mathbb{E}\|\nabla F(\psi_r)\|^2 \leq \mathbb{E}[F(\psi_r)] - \mathbb{E}[F(\psi_{r+1})] + \Gamma_2. \tag{55}$$

Summing over $r = 0, \ldots, R-1$, we have

$$\sum_{r=0}^{R-1}\Gamma_1\mathbb{E}\|\nabla F(\psi_r)\|^2 \leq \sum_{r=0}^{R-1}(\mathbb{E}[F(\psi_r)] - \mathbb{E}[F(\psi_{r+1})]) + \sum_{r=0}^{R-1}\Gamma_2$$
$$= F(\psi_0) - \mathbb{E}[F(\theta_R)] + R\Gamma_2$$
$$\leq F(\psi_0) - F^* + R\Gamma_2, \tag{56}$$

where $F^*$ is the minimum value of the global objective. Dividing by $R\Gamma_1$ gives the final result

$$\frac{1}{R}\sum_{r=0}^{R-1}\mathbb{E}\|\nabla F(\psi_r)\|^2 \leq \frac{F(\psi_0) - F^*}{R\Gamma_1} + \frac{\Gamma_2}{\Gamma_1}. \tag{57}$$

This completes the proof.

$\square$

## C ADDITIONAL RESULTS

### C.1 SENSITIVITY TO HYPERPARAMETERS

We now continue the analysis of sensitivity to key federated learning hyperparameters, most of the experiments are conducted under cross-device setting on CIFAR-10 and CIFAR-100. **First**, regarding the client participation rate, as shown in Figure 4, performance in the cross-silo setting remained stable with only a marginal drop at lower rates. In the cross-device setting, while all methods experience performance degradation with lower participation rates, FedPAC exhibited the highest resilience. **Furthermore**, we assessed the impact of the total number of clients and plot the result in Figure 5. It can be viewed that FedU2 achieves highest accuracy in settings with fewer clients, while FedPAC demonstrates superior robustness, maintaining more stable performance as the total number of clients changes. **Finally**, we analyze the effect of local training epochs in Figure 6. A reduction in local epochs led to significant performance degradation for all methods. In contrast, FedPAC maintained high accuracy across different epoch settings and could achieve a comparable level of accuracy to competing methods but with a reduced number of local epochs. **Collectively**, these experiments demonstrate the robustness of FedPAC to variations in the training configuration.

### C.2 ADDITIONAL TYPES OF DATA HETEROGENEITY

To further evaluate the robustness of FedPAC under diverse non-IID scenarios, we conducted additional experiments covering both **pathological label skew** and **domain shift**.

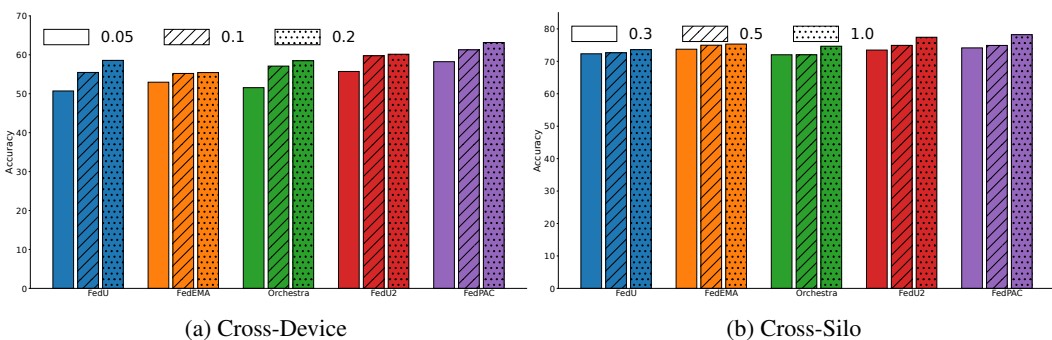

(a) Cross-Device            (b) Cross-Silo

Figure 4: Sensitivity to participation ratio on cross-device (left) and cross-silo (right) settings. Fed-PAC maintains stable accuracy even with low participation rates.

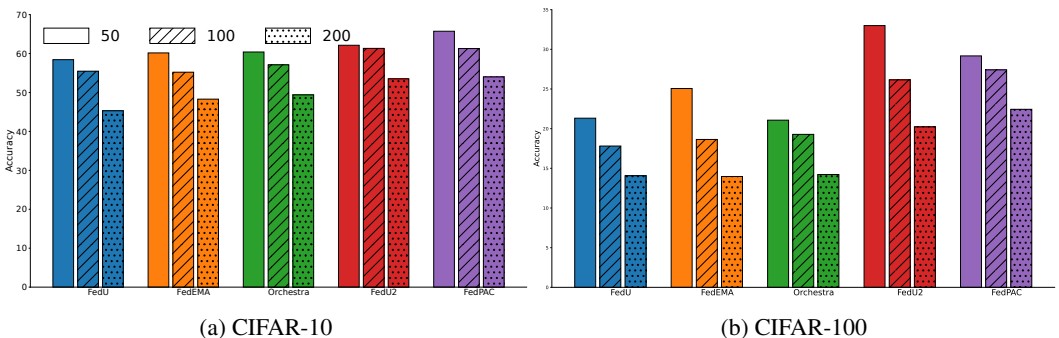

(a) CIFAR-10            (b) CIFAR-100

Figure 5: Sensitivity to clients number on CIFAR-10 (left) and CIFAR-100 (right) settings.

**Pathological Non-IID Settings.** While the main paper focuses on label heterogeneity via the Dirichlet distribution, we extend our evaluation to the **pathological non-IID setting**. In this scenario, each client holds data from a fixed number of classes. We conduct experiments on CIFAR-10 under both cross-silo and cross-device settings, where each client holds samples from only **2** or **4** distinct classes. As shown in Table 6, while BYOL maintains reasonable performance in the cross-silo setting, its accuracy deteriorates sharply in the cross-device setting. This collapse highlights the difficulty of learning generalizable features when local data becomes extremely sparse and fragmented. In contrast, FedPAC consistently outperforms baselines across all settings. This validates that our prototype-anchored consensus mechanism effectively incorporates global semantic information, preventing models overfitting to partial classes.

Table 6: Linear probing accuracy (%) on CIFAR-10 under pathological non-IID settings.

| Method | Cross-Device | | Cross-Silo | |
|---|---|---|---|---|
| | 2 Classes | 4 Classes | 2 Classes | 4 Classes |
| BYOL | 56.90 | 50.84 | 73.95 | 75.82 |
| FedX | 65.49 | 60.45 | 77.49 | 79.80 |
| FedU2[BYOL] | 67.43 | 65.49 | 82.26 | 83.17 |
| SSD | 66.12 | 63.73 | 82.59 | 84.48 |
| **FedPAC** | **70.81** | **66.48** | **85.04** | **86.44** |

**Domain Shift Scenarios.** To assess robustness against **feature distribution shifts** where clients hold data with different visual styles, we conducted experiments on the Digits-DG benchmark (Li et al., 2018), comprising three distinct datasets: MNIST (LeCun et al., 2002), SVHN (Netzer et al., 2011) and USPS (Hull, 2002). The experiment involved $N = 6$ clients, where each dataset is

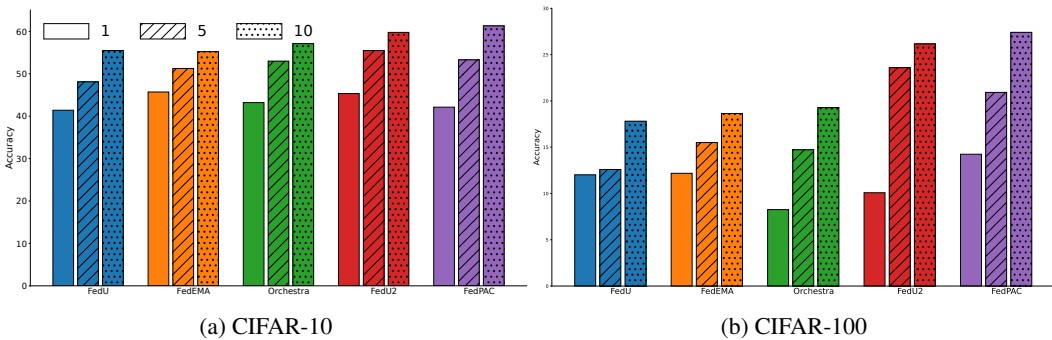

|            | (a) CIFAR-10 |            | (b) CIFAR-100 |

Figure 6: Sensitivity to local epochs on CIFAR-10 (left) and CIFAR-100 (right). FedPAC shows consistent robustness and efficiency across all settings.

assigned to two clients. All images were standardized to $32 \times 32$ with 3 channels to simulate a realistic scenario where clients collect semantically similar data from drastically different domains. As shown in Table 7, despite significant domain gaps, FedPAC successfully aligns representations into a unified semantic space. Specifically, FedPAC achieves superior accuracy on MNIST and USPS compared to state-of-the-art methods. While performance on SVHN is slightly lower, crucially, our method achieves the highest average accuracy across all datasets. This demonstrates that FedPAC is robust not only to label skew but also to severe feature shifts.

To further evaluate the robustness against severe distribution shifts, we extended our experiments to the DomainNet benchmark (Peng et al., 2019), which presents a more challenging feature shift scenario with six distinct domains. Specifically, we constructed a subset by selecting 10 common categories across all domains following existing works (Zhang et al., 2023; Chen et al., 2025). As reported in Table 8, we observe significant performance variance among baseline methods across different domains. Methods that excel in specific domains often struggle in others, suggesting they may overfit to specific domain styles rather than learning generalizable features. In contrast, Fed-PAC demonstrates superior stability and robustness. It ranks within the top-2 in five out of the six domains and achieves the highest average accuracy. Notably, FedPAC outperforms baselines by a substantial margin on abstract domains like **Quickdraw** and **Clipart**. These results indicate that our prototype-anchored consensus effectively reduces reliance on superficial domain-specific statistics. Instead, it encourages the model to capture semantics shared across heterogeneous clients, thereby demonstrating the strong capability in handling significant domain shifts.

Table 7: Linear probing and fine-tuning accuracy (%) on the Digits benchmark (MNIST, SVHN, USPS) under domain shift.

| Method | MNIST | | SVHN | | USPS | | Average | |
|---|---|---|---|---|---|---|---|---|
| | Linear | 1% | Linear | 1% | Linear | 1% | Linear | 1% |
| BYOL | 96.93 | 94.78 | 79.77 | 75.98 | 94.31 | 85.07 | 90.34 | 85.28 |
| FedX | 97.77 | 95.33 | 83.09 | 77.82 | 95.56 | 88.99 | 92.14 | 87.38 |
| FedU2$^{\text{BYOL}}$ | 99.01 | 97.33 | **87.25** | **81.48** | 96.51 | 92.32 | 94.26 | 90.38 |
| SSD | 98.67 | 97.57 | 86.18 | 80.29 | 96.92 | 89.65 | 93.92 | 89.17 |
| **FedPAC** | **99.13** | **97.87** | 86.24 | 79.86 | **97.46** | **93.57** | **94.28** | **90.43** |

## C.3 SCALABILITY TO LARGER MODEL ARCHITECTURE

While our primary evaluation employs ResNet-18, we extend our analysis to a deeper **ResNet-50** (He et al., 2016) backbone to verify that our method's superiority generalizes across architectures. We conducted experiments on CIFAR-10 under both cross-silo and cross-device settings. Table 9 summarizes the linear probing and fine-tuning accuracy. A key observation is that simply scaling up the model depth does not guarantee performance gains for all methods. Baselines such as BYOL

Table 8: Linear probing accuracy (%) on the DomainNet benchmark.

| Method | Clipart | Infograph | Painting | Quickdraw | Real | Sketch | Average |
|---|---|---|---|---|---|---|---|
| BYOL | 63.64 | 34.45 | 50.59 | 67.79 | 79.33 | 66.25 | 60.34 |
| FedX | 68.69 | 32.31 | 57.14 | 73.87 | 83.34 | 71.88 | 64.55 |
| FedU2$^{\text{BYOL}}$ | 67.83 | 35.25 | 61.26 | 76.34 | 86.10 | 74.03 | 66.80 |
| SSD | 66.95 | 44.01 | 69.75 | 82.53 | 69.10 | 69.97 | 67.05 |
| FedPAC | 74.19 | 37.06 | 62.62 | 86.47 | 73.43 | 73.12 | 67.82 |

and FedX exhibit instability, suffering from overfitting or training collapse, particularly in the challenging cross-device setting. In contrast, FedPAC achieves robust convergence and high accuracy, maintaining its superiority even with the deeper ResNet-50 backbone.

Table 9: Linear probing and fine-tuning accuracy (%) on CIFAR-10 using ResNet-50 backbone.

| Method | Cross-device | | | Cross-silo | | |
|---|---|---|---|---|---|---|
| | Linear | 1% | 10% | Linear | 1% | 10% |
| BYOL | 51.05 | 48.08 | 62.83 | 78.82 | 73.01 | 82.8 |
| FedX | 54.72 | 54.81 | 67.58 | 80.50 | 70.05 | 82.95 |
| FedU2$^{\text{BYOL}}$ | 70.56 | 57.03 | 73.37 | 84.22 | 72.25 | 84.90 |
| SSD | 72.73 | 63.08 | 76.39 | 85.60 | 72.92 | 84.85 |
| FedPAC | 75.01 | 64.41 | 78.01 | 88.30 | 77.93 | 86.49 |

## C.4 SCALABILITY TO DIFFERENT DATASETS

To assess scalability to datasets with more categories and higher resolution, we evaluated FedPAC on **Tiny-ImageNet** benchmark (Deng et al., 2009). We adopted a Cross-Silo setting with $N = 10$ clients under a non-IID partition ($\alpha = 0.1$). This setting is significantly more challenging than CIFAR-10/100 due to the larger number of semantic categories and the finer-grained visual distinctions. As shown in Table 10, FedPAC consistently outperforms baselines on the Tiny-ImageNet. Notably, the performance gap is more pronounced than in the CIFAR benchmarks. This suggests that as the semantic complexity of the dataset increases, the necessity of explicit semantic alignment becomes increasingly critical for learning discriminative representations.

Table 10: Linear probing and fine-tuning accuracy (%) on Tiny-ImageNet (Cross-Silo, $\alpha = 0.1$).

| Method | Linear | 1% | 10% |
|---|---|---|---|
| BYOL | 36.78 | 11.20 | 27.51 |
| FedX | 39.50 | 13.37 | 32.71 |
| FedU2$^{\text{BYOL}}$ | 43.74 | 15.65 | 34.22 |
| SSD | 42.42 | 13.78 | 32.84 |
| FedPAC | 47.44 | 20.28 | 38.20 |

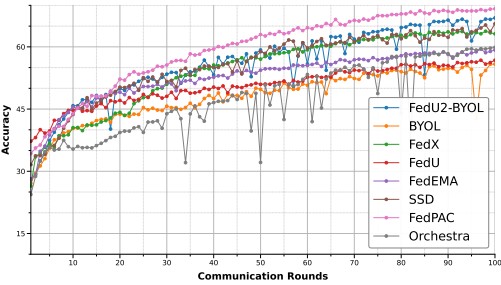

Figure 7: Convergence curve of KNN accuracy versus communication rounds on CIFAR-10.

## C.5 COMMUNICATION EFFICIENCY

Figure 7 illustrates the evolution of KNN accuracy across communication rounds for various methods on CIFAR-10 dataset under cross-device setting. FedPAC demonstrates higher communication

efficiency through rapid convergence, and it requires fewer rounds to attain the target accuracy compared to baselines, rendering it well-suited for resource-constrained FL scenarios.

## C.6 DATA AUGMENTATION

We employ a uniform augmentation strategy consistent with standard SSL protocols (Chen et al., 2020; Grill et al., 2020; Chen & He, 2021), applying transformations such as random cropping, color jittering, and horizontal flipping. To validate that this strategy does not introduce detrimental semantic shifts, we visualize the augmented samples in Figure 8. As observed, although the augmentations introduce pixel-level perturbations and visual diversity, the key structural and semantic features remain intact. This visualization supports our premise that a uniform augmentation strategy is feasible for federated unsupervised learning, as the semantic consistency is maintained across diverse local views.

## C.7 REPRESENTATION VISUALIZATION

To provide an intuitive and qualitative assessment of the learned representations, we visualize the feature embeddings using t-SNE. We first examine the problem of semantic misalignment by comparing local and global models from both FedU2 and FedPAC. As shown in Figure 9, while local models in FedU2 learn locally coherent representations, their feature spaces are misaligned with one another. Consequently, in the aggregated global model's feature space, representations from different categories may become mixed and thus reduce discriminability. In contrast, FedPAC learns maintains semantic consistency across clients, forming a global representation space with enhanced inter-class separation, demonstrating its ability to mitigate representation drift. Then we compare the final global representations learned by all methods in Figure 10. This visualization shows that FedPAC learns representations with better intra-class compactness and inter-class separability, providing further evidence of the high discriminative power of the feature space cultivated by our framework.

## D THE USE OF LARGE LANGUAGE MODELS (LLMs)

During the writing process, we used LLMs to identify grammatical errors in the article and polish some sentences.

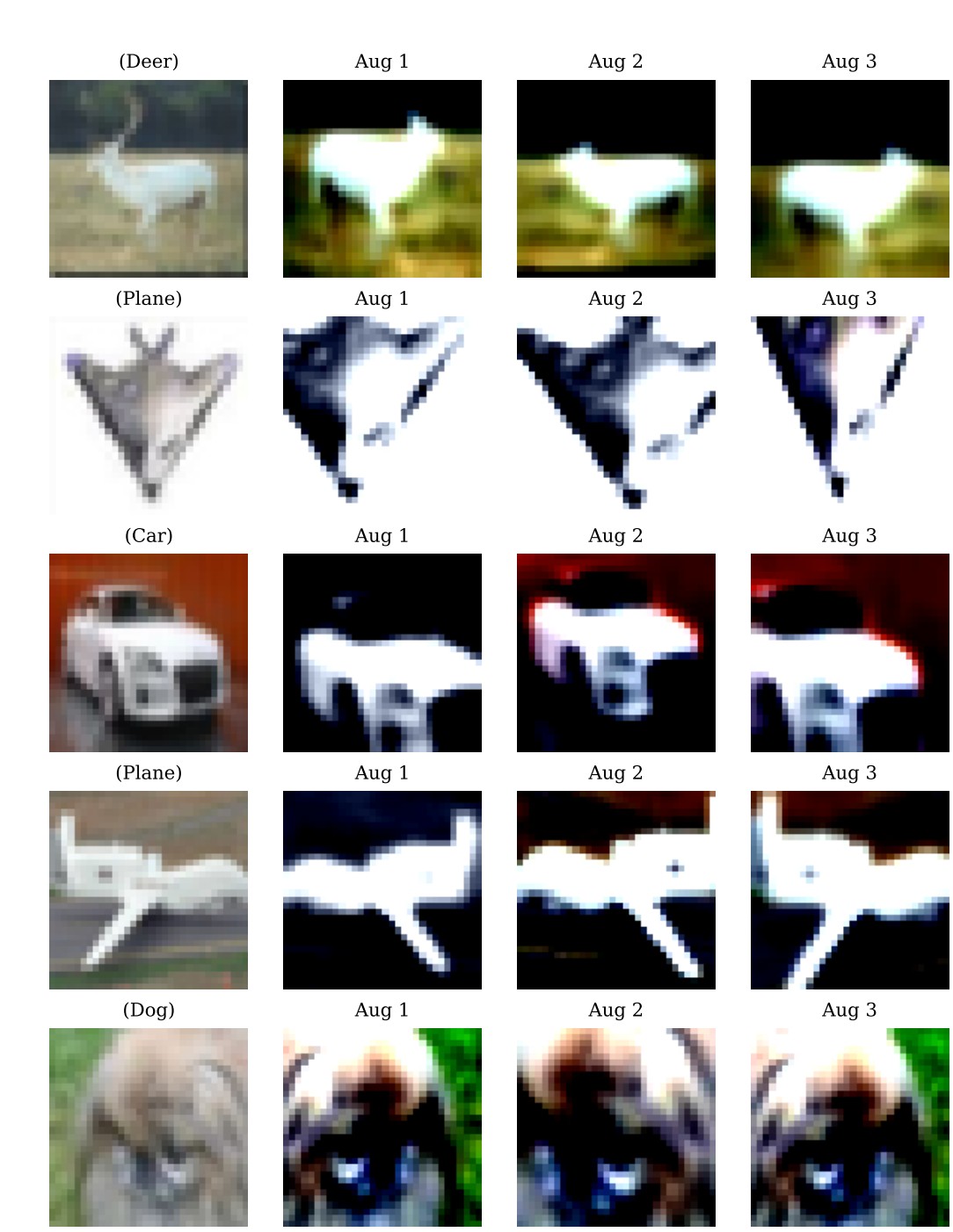

Figure 8: Visualization of augmented samples. The first row displays the original images, while the subsequent rows show the views generated by the uniform augmentation strategy. Despite the visual variations, the semantic identity of each sample is preserved.

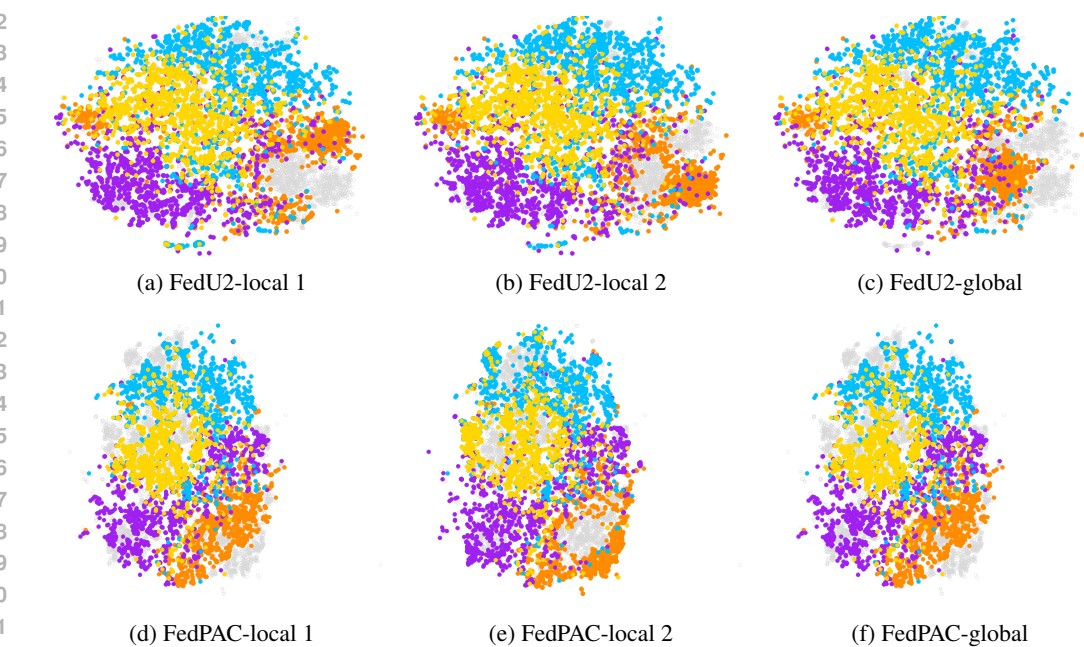

(a) FedU2-local 1      (b) FedU2-local 2      (c) FedU2-global

(d) FedPAC-local 1      (e) FedPAC-local 2      (f) FedPAC-global

Figure 9: Visualization of local and global representations, demonstrating that FedPAC alleviates cross-client representation drift, leading to enhanced global consistency compared to the FedU2.

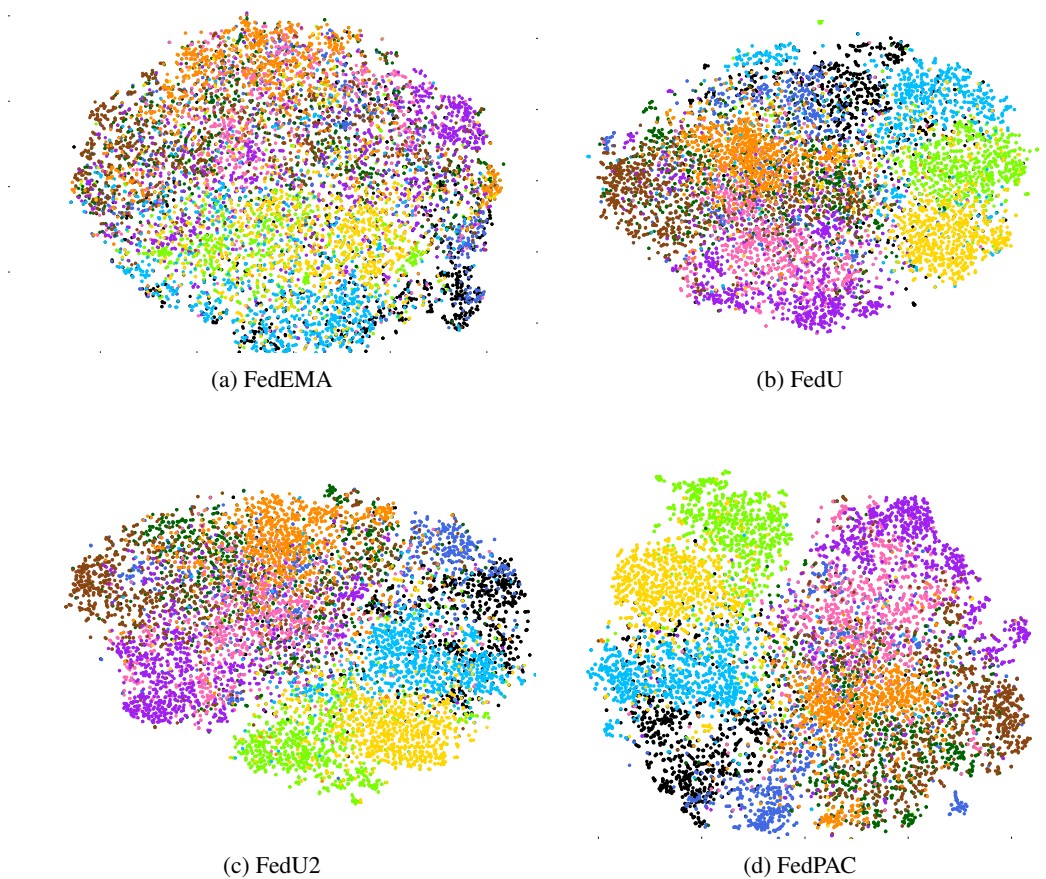

(a) FedEMA      (b) FedU

(c) FedU2      (d) FedPAC

Figure 10: Visualization of the representation space learned by the global model of different methods.

