# OpenReview forum: "FedPAC: Consistent Representation Learning for Federated Unsupervised Learning under Data Heterogeneity"
_ICLR.cc/2026/Conference — Submitted to ICLR 2026_

### Official Review · Reviewer_9cTj · 2025-10-23

**Soundness:** 3
**Presentation:** 2
**Contribution:** 3
**Rating:** 4
**Confidence:** 5

**Summary:**

This work proposes FedPAC to address the critical challenges of representation collapse and semantic misalignment in federated unsupervised learning. FedPAC introduces a dual-alignment objective during local training: a semantic alignment loss that steers local models towards a prototype-anchored consensus to ensure cross-client semantic consistency, coupled with a representation alignment loss that promotes the learning of discriminative and stable features. On the server, prototypes are aggregated by an optimization-based strategy that preserves semantic knowledge and ensure the prototypes remain representative. Extensive experiments CIFAR-10 and CIFAR-100 demonstrate the effectiveneses of proposed method.

**Strengths:**

1. This work proposes a synergistic architecture that combines a dual-alignment learning objective for clients’ local unsupervised learning and a prototype aggregation strategy refining global prototypes on the server.
2. This work provides a convergence analysis of proposed method.
3. Extensive experiments on CIFAR-10 and CIFAR-100 show the effectiveness of proposed method.

**Weaknesses:**

1. Many existing works explore the challenges the authors claim to have solved. What are the main distinctions between the proposed method and existing federated unsupervised learning methods? It is suggested that the authors add a comparative table to directly illustrate these distinctions. And it is recommended that the authors incorporate more recent and state-of-the-art works into Section 2.2.
2. In the convergence analysis, the definition of the prototype parameters $\rho$ conflicts with the strength/regularization parameter $\rho$ used in line 243. Furthermore, the prototype parameters appear to be updated using an SGD-like approach in every communication round. This requires a more detailed explanation of the update mechanism.
3. Is Assumption 6 commonly used in related literature? What is the practical meaning and implication of this assumption?
4. In Section 4.2.1, line 234, the authors claim that Unbalanced Optimal Transport (UOT) can accommodate scenarios where clients lack certain classes. However, there is no experimental evidence to support this claim. It is suggested that the authors add an experiment demonstrating performance under a pathological non-IID distribution.
5. In line 305, why is the number of local client prototypes (I) typically larger than the number of global prototypes (K), i.e., $I\geq K$?
6. The rigor of the presentation can be improved. Line 239 states that Equation (2) can be solved using Sinkhorn iterations, while line 247 refers to using a Sinkhorn-like iteration. The authors must use consistent and precise terminology to describe the solution method.

**Questions:**

See weaknesses.

---

> ### Author Response · Authors · 2025-11-26
> **Distinctions and Literature Update (Response to W1)**
>
> We thank the reviewer for the insightful comments, particularly regarding the theoretical analysis and the rigor of our presentation. We have carefully addressed each point as follows.
>
> We agree that clarifying the distinctions between FedPAC and existing methods is crucial. We have added a comparative table to directly illustrate the distinctions between FedPAC and existing federated unsupervised learning methods. Due to space constraints, we included the table in **Appendix A.4**. We have also updated **Section 2** to include the most recent state-of-the-art works to provide a more holistic and up-to-date overview of the field.

---

> ### Author Response · Authors · 2025-11-26
> **Notation and Update Mechanism (Response to W2 & W5)**
>
> **Notation Conflict Resolution.**
> We sincerely apologize for the notation conflict. In the revised manuscript, we have renamed the regularization parameter in the UOT objective from $\rho$ to $\mu$ to clearly distinguish it from the prototype parameters $\rho$ used in the convergence analysis. We have carefully proofread the paper to ensure consistent notation throughout.
>
> **Clarification on Prototype Aggregation.**
> We use $\mathbf{P}_l \in \mathbb{R}^{I \times d}$ to represent the collection of local prototypes uploaded by all participating clients in the current round. Specifically, if we have $K$ global prototypes and $M$ clients participating in training, the total number of local prototypes is $I = M \times K$. Since we typically have multiple participating clients ($M > 1$), it naturally follows that $I > K$.
>
> **Server-side Update Mechanism.**
> In the aggregation phase, we treat the global prototypes $\mathbf{P}_g$ as learnable parameters and the collected local prototypes $\mathbf{P}_l$ as training data. The objective is to minimize the prototype aggregation loss in **Section 4.3**. The server employs an SGD optimizer to perform $U$ epochs of iterative updates on $\mathbf{P}_g$. We have revised **Section 4.3** to explicitly describe this optimization process.

---

> ### Author Response · Authors · 2025-11-26
> **Validity and Implication of Assumption 6 (Response to W3)**
>
> Assumption 6 is analogous to the classical **Bounded Variance Assumption** for stochastic gradients, which is a standard staple in the convergence analysis of federated learning. In our setting, the server-side gradient $\nabla F_s$ acts as a stochastic estimator for the true global gradient $\nabla_\rho F$. Assumption 6 states that the variance of this estimation is bounded, and the parameter $\zeta^2$ quantifies the discrepancy between the server-side aggregation and the global optimization goal, which arises from data heterogeneity and partial client participation. This assumption guarantees that, although the update direction derived from the surrogate function $F_s$ is not identical to the true global gradient, the error is bounded. This guarantees the alignment between the surrogate optimization and the true global objective, ensuring that minimizing the surrogate function translates to a sufficient decrease in the global loss, thereby guaranteeing effective convergence.

---

> ### Author Response · Authors · 2025-11-26
> **Effectiveness of UOT on Missing Classes (Response to W4)**
>
> **Empirical Validation under Pathological Non-IID Settings.**
> To provide empirical evidence that Unbalanced Optimal Transport (UOT) can accommodate scenarios where clients lack certain classes, we conducted experiments under the **pathological non-IID setting**. We conducted experiments on CIFAR-10 under both cross-silo and cross-device settings, where each client holds samples from only **2** or **4** distinct classes. As shown in the tables below, FedPAC consistently outperforms competitive baselines.
>
> *Table 1: Performance comparison in a cross-device setting with 2 classes per client.*
> | Method | Linear | 1% | 10% |
> | :---: | :---: | :---: | :---: |
> | BYOL | 56.90  | 48.70  | 66.34 |
> | FedX | 65.49 | 54.62 | 73.47 |
> | FedU2-BYOL | 67.43|57.74|75.52|
> | SSD | 66.12 | 56.59 | 75.13 |
> | **FedPAC** | **70.81** | **60.94** | **77.42** |
>
>
>
> *Table 2: Performance comparison in a cross-silo setting with 2 classes per client.*
> | Method | Linear | 1% | 10% |
> | :---: | :---: | :---: | :---: |
> | BYOL | 73.95 | 67.61 | 78.33 |
> | FedX | 77.49 | 72.83 | 79.52 |
> | FedU2-BYOL | 82.26 | 74.55 | 82.06 |
> | SSD | 82.59 | 75.31 | 83.70 |
> | **FedPAC** | **85.04** | **75.60** | **85.50** |
>
>
>
> *Table 3: Performance comparison in a cross-device setting with 4 classes per client.*
> | Method | Linear | 1% | 10% |
> | :---: | :---: | :---: | :---: |
> | BYOL | 50.84 | 45.67 | 61.74 |
> | FedX | 60.45 | 52.83 | 69.35 |
> | FedU2-BYOL | 65.49|55.92|71.84|
> | SSD | 63.73 | 54.05 | 70.96 |
> | **FedPAC** | **66.48** | **57.49** | **74.40** |
>
>
>
> *Table 4: Performance comparison in a cross-silo setting with 4 classes per client.*
> | Method | Linear | 1% | 10% |
> | :---: | :---: | :---: | :---: |
> | BYOL | 75.82 | 70.23 | 80.57 |
> | FedX | 79.80  | 72.56 | 81.08 |
> | FedU2-BYOL | 83.17 | 74.17 | 83.20 |
> | SSD | 84.48 | 75.07 | 83.75 |
> | **FedPAC** | **86.44** | **79.82** | **86.83** |
>
>
> These results confirm that our UOT mechanism effectively handles missing classes by relaxing marginal constraints, maintaining high accuracy even when local data coverage is extremely sparse.

---

> ### Author Response · Authors · 2025-11-26
> **Terminological Rigor (Response to W6)**
>
> We thank the reviewer for pointing out the inconsistency in our terminology. We have revised the manuscript to use the precise term **generalized Sinkhorn iterations** consistently. In **Appendix A.1**, we have further elaborated on the derivation of this algorithm, detailing the semi-relaxed constraints used in our heterogeneous setting. In contrast, for server-side prototype aggregation, we employ standard Balanced OT, which can be solved via Sinkhorn-Knopp iterations [1].
>
> [1] Marco Cuturi. Sinkhorn distances: Lightspeed computation of optimal transport. Advances in neural information processing systems, 26, 2013.

---

> ### Author Response · Authors · 2025-11-30
> **Summary of Revisions**
>
> We appreciate the reviewer’s evaluation and the constructive comments. In response, we have incorporated:
>
> - Clarified distinctions (**W1**): Added a comparative table to explicitly illustrate the distinctions between our approach and existing federated unsupervised learning methods (**Appendix A.4**).
> - Expanded references (**W1**): Incorporated more recent works to provide a more holistic and up-to-date overview of the field (**Section 2.2 & 2.3**).
> - Methodological Clarification (**W2, W3, W5**): Elucidated the rationale behind Assumption 6 and provided detailed explanations for the server-side prototype aggregation mechanism (**Section 4.3**). Additionally, we standardized terminology throughout the manuscript to ensure rigor.
> - Verified UOT Mechanism (**W4**): Conducted experiments on the pathological Non-IID setting, empirically proving that UOT effectively handles scenarios where clients lack certain classes (**Appendix C.2**).
>
> Thank you again for your review and the opportunity to improve our work.

---

### Official Review · Reviewer_NVU7 · 2025-10-31

**Soundness:** 2
**Presentation:** 2
**Contribution:** 3
**Rating:** 4
**Confidence:** 3

**Summary:**

This paper proposes FedPAC, a novel federated unsupervised learning framework designed to address two critical challenges under data heterogeneity: representation collapse and semantic misalignment. FedPAC introduces a dual-alignment objective on the client side, including a semantic alignment loss (enforcing consistency with global prototypes) and a representation alignment loss (promoting discriminative and stable features). On the server side, a prototype optimization strategy refines global prototypes to preserve semantic diversity and maintain cross-client coherence. The paper provides theoretical convergence guarantees and extensive experiments on CIFAR-10 and CIFAR-100, showing clear improvements over prior federated unsupervised learning methods.

**Strengths:**

1. The paper effectively integrates semantic alignment (with global prototypes) and representation alignment (across augmented views), encouraging clients to learn representations that are both locally discriminative and globally consistent. This dual-loss design directly addresses data heterogeneity in a principled manner.

2. By optimizing global prototypes on the server, FedPAC maintains prototype diversity and captures the distribution of client representations more effectively. This prototype-centered aggregation approach is rarely explored in prior federated unsupervised learning literature and contributes a fresh perspective on maintaining global semantic consistency.

3. The paper provides a detailed convergence analysis proving sufficient decrease and global convergence bounds under mild assumptions.

**Weaknesses:**

1. Experiments are conducted only on CIFAR-10 and CIFAR-100 using Dirichlet sampling to simulate data heterogeneity. The evaluation could be strengthened by including additional datasets (e.g., MNIST, Tiny-ImageNet, or DomainNet) and broader forms of heterogeneity (e.g., domain shift, quantity-based label imbalance). This would help assess the generalization of FedPAC in more realistic federated scenarios.

2. The paper does not examine how FedPAC performs under varying numbers of clients, client participation ratios, or communication budgets. These factors are crucial in practical federated environments and could significantly impact convergence and accuracy.

3. There are small typographical errors and redundancies (e.g., repeated “sample sample” at line 225). A careful proofreading pass would improve the overall presentation quality.

**Questions:**

1. Can the proposed method maintain its performance advantage under other types of data heterogeneity, such as domain shift scenarios where the data distribution varies in style or modality across clients, rather than only under label distribution heterogeneity (i.e., Dirichlet-based label shift)?

2. The experiments only use ResNet-18 as the encoder. Would the method still hold its superiority when using different backbones such as ResNet-50, ViT, or MobileNet, particularly given their different representational capacities?

---

> ### Author Response · Authors · 2025-11-26
> **Evaluation on Broader Forms of Heterogeneity (Response to W1 & Q1,2)**
>
> We sincerely thank the reviewer for the constructive feedback and for acknowledging the potential of our work. We have conducted extensive additional experiments to address the concerns regarding generalization, robustness, and scalability.
>
> **Performance under Pathological Non-IID Settings (Label Skew).**
> To address the suggestion for broader heterogeneity beyond Dirichlet sampling, we extended our evaluation to the **pathological non-IID setting**. In this scenario, each client holds data from a fixed number of classes. We conducted experiments on CIFAR-10 under both cross-silo and cross-device settings, where each client holds samples from only **2** or **4** distinct classes. For these extensive additional experiments, we compared FedPAC against competitive baselines, including the recent SOTA method **SSD (ICCV 2025)** [1].
>
> As shown in the tables below, FedPAC consistently outperforms baselines in both Cross-Device and Cross-Silo settings, demonstrating robust generalization under severe label skew.
>
> *Table1: Cross-device & 2 classes.*
> | Method | Linear | 1% | 10% |
> | :---: | :---: | :---: | :---: |
> | BYOL | 56.90  | 48.70  | 66.34 |
> | FedX | 65.49 | 54.62 | 73.47 |
> | FedU2-BYOL | 67.43|57.74|75.52|
> | SSD | 66.12 | 56.59 | 75.13 |
> | **FedPAC** | **70.81** | **60.94** | **77.42** |
>
>
>
> *Table2: Cross-silo & 2 classes.*
> | Method | Linear | 1% | 10% |
> | :---: | :---: | :---: | :---: |
> | BYOL | 73.95 | 67.61 | 78.33 |
> | FedX | 77.49 | 72.83 | 79.52 |
> | FedU2-BYOL | 82.26 | 74.55 | 82.06 |
> | SSD | 82.59 | 75.31 | 83.70 |
> | **FedPAC** | **85.04** | **75.60** | **85.50** |
>
>
>
> *Table3: Cross-device & 4 classes.*
> | Method | Linear | 1% | 10% |
> | :---: | :---: | :---: | :---: |
> | BYOL | 50.84 | 45.67 | 61.74 |
> | FedX | 60.45 | 52.83 | 69.35 |
> | FedU2-BYOL | 65.49|55.92|71.84|
> | SSD | 63.73 | 54.05 | 70.96 |
> | **FedPAC** | **66.48** | **57.49** | **74.40** |
>
>
>
> *Table4: Cross-silo & 4 classes.*
> | Method | Linear | 1% | 10% |
> | :---: | :---: | :---: | :---: |
> | BYOL | 75.82 | 70.23 | 80.57 |
> | FedX | 79.80  | 72.56 | 81.08 |
> | FedU2-BYOL | 83.17 | 74.17 | 83.20 |
> | SSD | 84.48 | 75.07 | 83.75 |
> | **FedPAC** | **86.44** | **79.82** | **86.83** |
>
> **Robustness to Feature Distribution Shifts (Domain Shift).**
> To answer the question regarding robustness against feature distribution shifts, we conducted experiments on the **Digits-DG benchmark** [2] (including MNIST [3], SVHN [4], USPS [5]). We set up the experiments with $N=6$ clients, where each domain is assigned to two clients. This simulates a realistic scenario with severe feature distribution shifts.
>
> The results below show that FedPAC achieves the highest average accuracy. Notably, it demonstrates superior performance on MNIST and USPS compared to SOTA methods, proving its effectiveness in handling domain shifts.
>
> *Table 5: Linear probing accuracy.*
> | Method | MNIST | SVHN | USPS | AVG |
> | :---: | :---: | :---: | :---: | :---: |
> | BYOL | 96.93 | 79.77 | 94.31 | 90.34 |
> | FedX | 97.77  | 83.09 | 95.56 | 92.14 |
> | FedU2-BYOL | 99.01 | **87.25** | 96.51 | 94.26 |
> | SSD | 98.67 | 86.18 | 96.92 | 93.92 |
> | **FedPAC** | **99.13** | 86.24 | **97.46** | **94.28** |
>
>
>
> *Table 6: 1% fine-tuning accuracy.*
> | Method | MNIST | SVHN | USPS | AVG |
> | :---: | :---: | :---: | :---: | :---: |
> | BYOL | 94.78 | 75.98 | 85.07 | 85.28 |
> | FedX | 95.33 | 77.82 | 88.99 | 87.38 |
> | FedU2-BYOL | 97.33 | **81.48** | 92.32 | 90.38 |
> | SSD | 97.57 | 80.29 | 89.65 | 89.17 |
> | **FedPAC** | **97.87** | 79.86 | **93.57** | **90.43** |
>
>
>
> *Table 7: 10% fine-tuning accuracy.*
> | Method | MNIST | SVHN | USPS | AVG |
> | :---: | :---: | :---: | :---: | :---: |
> | BYOL | 97.96 | 85.77 | 94.01 | 92.58 |
> | FedX | 98.84 | 87.49 | 95.02 | 93.78 |
> | FedU2-BYOL | 98.87 | 89.56 | 96.41 | 94.95 |
> | SSD | 99.01 | **89.91** | 95.31 | 94.74 |
> | **FedPAC** | **99.10** | 89.78 | **96.86** | **95.25** |
>
> [1] Hung-Chieh Fang, Hsuan-Tien Lin, Irwin King, and Yifei Zhang. Soft separation and distillation: Toward global uniformity in federated unsupervised learning. In Proceedings of the IEEE/CVF International Conference on Computer Vision, pp. 2971–2980, 2025.
>
> [2] Ya Li, Xinmei Tian, Mingming Gong, Yajing Liu, Tongliang Liu, Kun Zhang, and Dacheng Tao. Deep domain generalization via conditional invariant adversarial networks. In Proceedings of the European conference on computer vision (ECCV), pp. 624–639, 2018.
>
> [3] Yann LeCun, L´eon Bottou, Yoshua Bengio, and Patrick Haffner. Gradient-based learning applied to document recognition. Proceedings of the IEEE, 86(11):2278–2324, 2002.
>
> [4] Yuval Netzer, Tao Wang, Adam Coates, Alessandro Bissacco, Baolin Wu, Andrew Y Ng, et al. Reading digits in natural images with unsupervised feature learning. In NIPS workshop on deep learning and unsupervised feature learning, volume 2011, pp. 7. Granada, 2011.
>
> [5] Jonathan J. Hull. A database for handwritten text recognition research. IEEE Transactions on pattern analysis and machine intelligence, 16(5):550–554, 2002.

---

> ### Author Response · Authors · 2025-11-26
> **Generalization and Scalability (Response to W1 & Q2)**
>
> **Scalability to Larger Model Architectures (ResNet-50).**
> To address **Q2**, we extended our analysis from ResNet-18 to the deeper **ResNet-50** [1] backbone. We conducted experiments on CIFAR-10 under both cross-silo and cross-device settings. The following tables summarize the accuracies.
>
> A key observation is that simply scaling up model depth causes instability for baselines like BYOL and FedX, e.g., overfitting or collapse under the cross-device setting. In contrast, FedPAC achieves robust convergence and maintains its superiority with the deeper ResNet-50 backbone, validating its scalability to larger architectures.
>
> *Table 1: Performance comparison in a cross-device setting with ResNet-50.*
> | Method | Linear | 1% | 10% |
> | :---: | :---: | :---: | :---: |
> | BYOL | 51.05 | 48.08 | 62.83 |
> | FedX | 54.72 | 54.81 | 67.58 |
> | FedU2-BYOL | 70.56 | 57.03 | 73.37 |
> | SSD | 72.73 | 63.08| 76.39 |
> | **FedPAC** | **75.01** | **64.41** | **78.01** |
>
>
>
> *Table 2: Performance comparison in a cross-silo setting with ResNet-50.*
> | Method | Linear | 1% | 10% |
> | :---: | :---: | :---: | :---: |
> | BYOL | 78.92 | 73.01 | 82.80 |
> | FedX | 80.50  | 70.05 | 82.95 |
> | FedU2-BYOL | 84.22 | 72.25 | 84.90 |
> | SSD | 85.60 | 72.92| 84.85 |
> | **FedPAC** | **88.30** | **77.93** | **86.49** |
>
>
> **Scalability to Larger Datasets (Tiny-ImageNet).**
> To assess scalability to datasets with more categories and higher resolution, we evaluated on the **Tiny-ImageNet** benchmark [2]. We adopted a Cross-Silo setting with $N=10$ clients under a non-IID partition ($\alpha=0.1$). As shown below, FedPAC consistently outperforms baselines on Tiny-ImageNet, with a more pronounced performance gap than in CIFAR benchmarks, highlighting its advantage in handling more complex datasets.
>
>
> *Table 3: Performance comparison on Tiny-ImageNet.*
> | Method | Linear | 1% | 10% |
> | :---: | :---: | :---: | :---: |
> | BYOL | 36.78 | 11.20 | 27.51 |
> | FedX | 39.50 | 13.37 | 32.71 |
> | FedU2-BYOL | 43.74 | 15.65| 34.22 |
> | SSD | 42.42 | 13.78| 32.84 |
> | **FedPAC** | **47.44** | **20.28** | **38.20** |
>
> [1] Kaiming He, Xiangyu Zhang, Shaoqing Ren, and Jian Sun. Deep residual learning for image recognition. In Proceedings of the IEEE conference on computer vision and pattern recognition, pp.770–778, 2016.
>
> [2] Jia Deng, Wei Dong, Richard Socher, Li-Jia Li, Kai Li, and Li Fei-Fei. Imagenet: A large-scale hierarchical image database. In 2009 IEEE conference on computer vision and pattern recognition, pp. 248–255. Ieee, 2009.

---

> ### Author Response · Authors · 2025-11-26
> **Robustness to Federated Hyperparameters (Response to W2)**
>
> We acknowledge the importance of practical federated settings. We respectfully clarify that we have conducted comprehensive sensitivity analyses regarding numbers of clients, client participation ratios and local epochs. Due to space constraints, the detailed plots and tables are presented in **Appendix C.1**. Results indicate that FedPAC remains robust across these variations.

---

> ### Author Response · Authors · 2025-11-26
> **Presentation Quality (Response to W3)**
>
> We have conducted a thorough proofreading of the entire paper and corrected the redundancy. We have fixed grammatical errors and refined sentence structures to ensure high presentation quality.

---

> ### Author Response · Authors · 2025-11-30
> **Additional Results on Robustness to domain Shift (Response to W1 & Q1)**
>
> To further evaluate the robustness against severe domain shifts, we extended our experiments to the DomainNet benchmark [1], which presents a more challenging feature shift scenario with six distinct domains. Specifically, we constructed a subset by selecting 10 common categories across all domains following existing works [2,3]. As reported in the table below, we observe significant performance variance among baseline methods across different domains. Methods that excel in specific domains often struggle in others, suggesting they may overfit to specific domain styles rather than learning generalizable features. In contrast, FedPAC demonstrates superior stability and robustness. It ranks within the top-2 in five out of the six domains and achieves the highest average accuracy. Notably, FedPAC outperforms baselines by a substantial margin on abstract domains like \textbf{Quickdraw} and \textbf{Clipart}. These results indicate that our prototype-anchored consensus effectively reduces reliance on superficial domain-specific statistics. Instead, it encourages the model to capture semantics shared across heterogeneous clients, thereby demonstrating the strong capability in handling significant domain shifts beyond simple label distribution heterogeneity.
>
> *Table 1: Linear probing accuracy on DomainNet-10.*
> | Method | Clipart | Infograph | Painting | Quickdraw | Real | Sketch | AVG |
> | :---: | :---: | :---: | :---: | :---: | :---: | :---: | :---: |
> | BYOL | 63.64 | 34.45 | 50.59 | 67.79 | 79.33 | 66.25 | 60.34 |
> | FedX | 68.69 | 32.31 | 57.14 | 73.87 | 83.34 | 71.88 | 64.55 |
> | FedU2-BYOL | 67.83 | 35.25 | 61.26 | 76.34 | **86.10** | **74.03** | 66.80 |
> | SSD | 66.95 | **44.01** | **69.75** | 82.53| 69.10 | 69.97 | 67.05 |
> | **FedPAC** | **74.19** | 37.06 | 62.62 | **86.47** | 73.43 | 73.12 | **67.82** |
>
> [1] Xingchao Peng, Qinxun Bai, Xide Xia, Zijun Huang, Kate Saenko, and Bo Wang. Moment matching for multi-source domain adaptation. In Proceedings of the IEEE/CVF international conference on computer vision, pp. 1406–1415, 2019.
>
> [2] Fengda Zhang, Kun Kuang, Long Chen, Zhaoyang You, Tao Shen, Jun Xiao, Yin Zhang, Chao Wu, Fei Wu, Yueting Zhuang, et al. Federated unsupervised representation learning. Frontiers of Information Technology & Electronic Engineering, 24(8):1181–1193, 2023.
>
> [3] Haokun Chen, Hang Li, Yao Zhang, Jinhe Bi, Gengyuan Zhang, Yueqi Zhang, Philip Torr, Jindong Gu, Denis Krompass, and Volker Tresp. Fedbip: Heterogeneous one-shot federated learning with personalized latent diffusion models. In Proceedings of the Computer Vision and Pattern Recognition Conference, pp. 30440–30450, 2025.

---

> ### Author Response · Authors · 2025-11-30
> **Summary of Revisions**
>
> We sincerely thank the reviewer for the insightful feedback and constructive suggestions. In response, we have included:
> - Broader forms of heterogeneity (**W1, Q1**): Extended our evaluation to pathological distributions and domain shift scenarios (**Appendix C.2**).
> - Different backbones (**W1, Q2**): Validated our method using a deeper ResNet-50 architecture (**Appendix C.3**).
> - Broader Datasets (**W1**): Extended our evaluation to the Tiny-ImageNet benchmark (**Appendix C.4**).
> - Federated learning settings (**W2**): Clarified the sensitivity analyses on client numbers, participation ratios, and local epochs (**Appendix C.1**).
>
> We appreciate your time and effort in evaluating our work. The comments have been instrumental in strengthening our manuscript.

---

### Official Review · Reviewer_p9Fz · 2025-10-31

**Soundness:** 3
**Presentation:** 3
**Contribution:** 3
**Rating:** 6
**Confidence:** 3

**Summary:**

This paper proposes a federated unsupervised learning framework named FedPAC, designed to address representation learning challenges under non-IID data distributions. FedPAC introduces a prototype-anchored consensus mechanism, incorporating dual alignment objectives during client-side local training. On one hand, the representation alignment loss ensures invariance across augmented views; on the other hand, the semantic alignment loss aligns local representations with globally shared prototypes, thereby achieving global semantic consistency. The server aggregates local prototypes via an optimization strategy to update the global prototypes, enhancing cross-client semantic consensus. Experimental results demonstrate that FedPAC outperforms existing methods on the CIFAR-10 and CIFAR-100 benchmark datasets, achieving superior performance.

**Strengths:**

(1) The FedPAC method integrates dual alignment objectives in client-side local unsupervised learning with a prototype aggregation strategy for server-side global prototype optimization, achieving a balance between local representations and global consistency.

(2) This paper conducts a mathematical analysis of FedPAC's convergence, theoretically demonstrating that the method achieves stable convergence within a finite number of communication rounds.

**Weaknesses:**

(1) The related work section cites studies that are not sufficiently recent. Additionally, the paper includes only 22 references, with just 3 published in the last three years (2023 and beyond), which does not provide readers with a comprehensive overview of the broader topic.

(2) The FedPAC employs the Unbalanced Optimal Transport (UOT) method to relax the prototype marginals, thereby preventing spurious assignments. In Section 3, FedPAC assumes that all clients utilize the same augmentation function $\mathcal{T}$, without accounting for the potential impacts of feature shift or noise on model performance.

(3) After completing local training on the clients, the model parameters and local prototypes are uploaded to the server, which may introduce privacy leakage risks. However, the paper does not provide sufficient discussion on this issue.

(4) The FedPAC introduces multiple hyperparameters (such as the entropy regularization parameter $\epsilon$, KL penalty strength $\rho$, loss balancing coefficient $\lambda$, uniformity trade-off coefficient $\beta$, and repulsion sharpness $\gamma$, among others). The paper merely describes their roles without providing specific values in the experimental setup.

(5) The paper lacks an analysis of the time complexity and space complexity of FedPAC, and does not provide a comparison of computational efficiency with baseline methods.

(6) The experimental section relies exclusively on CIFAR-10 and CIFAR-100 as benchmark datasets. Although these two datasets differ in class granularity, they are fundamentally similar in nature. It is advisable to use a broader set of datasets to enhance the persuasiveness of the experimental results.

(7) The baseline methods cited in the paper are somewhat outdated, with the most recent being from 2024 (FedU2), while the majority stem from 2022 or earlier. It is recommended to incorporate additional recent comparative methods for a more comprehensive evaluation.

**Questions:**

(1) The FedPAC method employs UOT to relax prototype marginals in order to prevent spurious assignments, yet it assumes that all clients use the same augmentation function $\mathcal{T}$. How does it address the potential impacts of feature shift or noise? What is the robustness of this mechanism under extreme non-IID scenarios?

(2) The process of clients uploading model parameters and local prototypes to the server may introduce privacy leakage risks. Have the authors considered incorporating protective measures?

(3) The paper lacks an analysis of FedPAC's time and space complexities, as well as a comparison of computational efficiency with baseline methods. How do the authors evaluate the method's practical deployability, particularly in resource-constrained federated environments?

(4) The majority of the baseline methods compared are from 2022 or earlier, with the most recent being FedU2 from 2024. Could the authors extend the comparisons to include more recent methods to enhance the comprehensiveness of the evaluation?

---

> ### Author Response · Authors · 2025-11-26
> **Literature and Baselines (Response to W1, W7 & Q4)**
>
> We sincerely thank the reviewer for the comprehensive evaluation and for pointing out the missing references and the need for broader experimental validation. We have addressed these concerns point-by-point below.
>
> We have substantially expanded our references including more works published from 2023 to 2025, providing a holistic overview of the field. And we have incorporated the very latest SOTA method, **SSD (ICCV 2025)** [1], into our comparative analysis. As detailed in the experiments, FedPAC consistently outperforms these baselines.
>
> [1] Hung-Chieh Fang, Hsuan-Tien Lin, Irwin King, and Yifei Zhang. Soft separation and distillation: Toward global uniformity in federated unsupervised learning. In Proceedings of the IEEE/CVF International Conference on Computer Vision, pp. 2971–2980, 2025.

---

> ### Author Response · Authors · 2025-11-26
> **Robustness of Augmentation and UOT (Response to W2 & Q1)**
>
> We employ a uniform augmentation strategy consistent with standard SSL protocols [1,2,3]. While tailoring augmentations to specific clients is theoretically appealing, it is often infeasible due to privacy constraints. To validate that this strategy does not introduce detrimental semantic shifts, we have added visualizations of augmented samples in **Appendix C.6** to confirm that the key structural and semantic features remain intact despite perturbations. This visualization supports our premise that a uniform augmentation strategy is feasible for federated unsupervised learning, as the semantic consistency is maintained across diverse local views.
>
> The reviewer correctly notes that uniform augmentation might introduce noise or feature shifts. However, our Unbalanced Optimal Transport (UOT) mechanism provides a natural defense against this. If an augmented sample is noisy or shifts significantly, and thus bears low similarity to existing prototypes, the entropy regularization in UOT forces its assignment to approach a uniform distribution. When computing the Cross-Entropy loss, such a uniform distribution generates ambiguous and weak gradient signals. This prevents the model from being forcefully updated in an incorrect direction, effectively neutralizing the impact of augmentation artifacts.
>
> To empirically validate the robustness of this mechanism under extreme non-IID scenarios, we refer to our additional experiments on the Pathological Non-IID setting, which represents a highly skewed distribution. We conducted experiments on CIFAR-10 under both cross-silo and cross-device settings, where each client holds samples from only **2** or **4** distinct classes. As shown in the tables below, FedPAC significantly outperforms baselines in this setting, confirming that our method remains robust and effective even under extreme non-IID scenarios.
>
> *Table 1: Performance comparison in a cross-device setting with 2 classes per client.*
>
> | Method | Linear | 1% | 10% |
> | :---: | :---: | :---: | :---: |
> | BYOL | 56.90  | 48.70  | 66.34 |
> | FedX | 65.49 | 54.62 | 73.47 |
> | FedU2-BYOL | 67.43|57.74|75.52|
> | SSD | 66.12 | 56.59 | 75.13 |
> | **FedPAC** | **70.81** | **60.94** | **77.42** |
>
>
> *Table 2: Performance comparison in a cross-silo setting with 2 classes per client.*
> | Method | Linear | 1% | 10% |
> | :---: | :---: | :---: | :---: |
> | BYOL | 73.95 | 67.61 | 78.33 |
> | FedX | 77.49 | 72.83 | 79.52 |
> | FedU2-BYOL | 82.26 | 74.55 | 82.06 |
> | SSD | 82.59 | 75.31 | 83.70 |
> | **FedPAC** | **85.04** | **75.60** | **85.50** |
>
>
>
> *Table 3: Performance comparison in a cross-device setting with 4 classes per client.*
> | Method | Linear | 1% | 10% |
> | :---: | :---: | :---: | :---: |
> | BYOL | 50.84 | 45.67 | 61.74 |
> | FedX | 60.45 | 52.83 | 69.35 |
> | FedU2-BYOL | 65.49|55.92|71.84|
> | SSD | 63.73 | 54.05 | 70.96 |
> | **FedPAC** | **66.48** | **57.49** | **74.40** |
>
>
>
> *Table 4: Performance comparison in a cross-silo setting with 4 classes per client.*
> | Method | Linear | 1% | 10% |
> | :---: | :---: | :---: | :---: |
> | BYOL | 75.82 | 70.23 | 80.57 |
> | FedX | 79.80  | 72.56 | 81.08 |
> | FedU2-BYOL | 83.17 | 74.17 | 83.20 |
> | SSD | 84.48 | 75.07 | 83.75 |
> | **FedPAC** | **86.44** | **79.82** | **86.83** |
>
> [1] Ting Chen, Simon Kornblith, Mohammad Norouzi, and Geoffrey Hinton. A simple framework for contrastive learning of visual representations. In International conference on machine learning pp. 1597–1607. PmLR, 2020.
>
> [2] Jean-Bastien Grill, Florian Strub, Florent Altch´e, Corentin Tallec, Pierre Richemond, Elena Buchatskaya, Carl Doersch, Bernardo Avila Pires, Zhaohan Guo, Mohammad Gheshlaghi Azar, et al. Bootstrap your own latent-a new approach to self-supervised learning. Advances in neural information processing systems, 33:21271–21284, 2020.
>
> [3] Xinlei Chen and Kaiming He. Exploring simple siamese representation learning. In Proceedings of the IEEE/CVF conference on computer vision and pattern recognition, pp. 15750–15758, 2021.

---

> ### Author Response · Authors · 2025-11-26
> **Privacy Considerations (Response to W3 & Q2)**
>
> While the primary objective of this work is learning representation in federated unsupervised learning, we recognize the concomitant privacy implications. FedPAC entails the transmission of both model parameters and local prototypes. While the exchange of model parameters is inherent to standard Federated Learning, prototypes serve as highly compressed statistical representations and the privacy risk associated with their transmission is significantly lower than sharing raw data. Furthermore, FedPAC is compatible with standard privacy mechanisms.  For instance, Differential Privacy (DP) [1] can be applied by injecting Gaussian or Laplacian noise into parameters and prototypes prior to transmission, and Secure Aggregation (SecAgg) [2] can be used to aggregate updates. We have added a dedicated discussion about privacy risks and mitigation strategies in **Appendix A.4**, outlining these integrations as a priority for future work.
>
> [1] Cynthia Dwork, Frank McSherry, Kobbi Nissim, and Adam Smith. Calibrating noise to sensitivity in private data analysis. In Theory of cryptography conference, pp. 265–284. Springer, 2006.
>
> [2] Keith Bonawitz, Vladimir Ivanov, Ben Kreuter, Antonio Marcedone, H Brendan McMahan, Sarvar Patel, Daniel Ramage, Aaron Segal, and Karn Seth. Practical secure aggregation for privacy- preserving machine learning. In proceedings of the 2017 ACM SIGSAC Conference on Computer and Communications Security, pp. 1175–1191, 2017.

---

> ### Author Response · Authors · 2025-11-26
> **Complexity Analysis (Response to W5 & Q3)**
>
> **Time and Space Complexity.**
> We address the concern regarding computational efficiency. Theoretically, the complexity of FedPAC is asymptotically dominated by the backbone network, similar to other methods. Specifically, regarding space complexity, the storage overhead for prototypes is negligible compared to the model parameters. Similarly, for time complexity, the additional overhead introduced by the UOT solver involves low-dimensional matrix operations, which is negligible compared to the computational cost of the backbone.
>
> Let $d$ denote the number of model parameters and $N$ denote the number of clients. The tables below compare the time and space complexities of FedPAC against other methods. As illustrated, both the space and time complexities of all methods remain at the $O(d)$ level.
>
> *Table 1: Comparison of time complexity.*
> | Method | Client | Server | Global |
> | :---: | :---: | :---: | :---: |
> | FedU | $O(2d)$ | $O(N \times d)$ | $O(d)$ |
> | FedX | $O(2d)$ | $O(N \times d)$ | $O(d)$ |
> | FedEMA | $O(d)$ | $O(N \times d)$ | $O(d)$ |
> | Orchestra | $O(2d)$ | $O(N \times d)$ | $O(d)$ |
> | FedU2 | $O(2d)$ | $O(N^2 \times d)$ | $O(d)$ |
> | SSD | $O(2d)$ | $O(N \times d)$ | $O(d)$ |
> | FedPAC | $O(2d)$ | $O(N \times d)$ | $O(d)$ |
>
>
>
> *Table 2: Comparison of space complexity.*
> | Method | Client | Server | Global |
> | :---: | :---: | :---: | :---: |
> | FedU | $O(2d)$ | $O(d)$ | $O(d)$ |
> | FedX | $O(2d)$ | $O(d)$ | $O(d)$ |
> | FedEMA | $O(d)$ | $O(2d)$ | $O(d)$ |
> | Orchestra | $O(2d)$ | $O(d)$ | $O(d)$ |
> | FedU2 | $O(2d)$ | $O(d)$ | $O(d)$ |
> | SSD | $O(d)$ | $O(d)$ | $O(d)$ |
> | FedPAC | $O(2d)$ | $O(d)$ | $O(d)$ |
>
>
> **Communication Efficiency.**
> Practically, FedPAC achieves higher efficiency through faster convergence. As shown in **Appendix C.5**, our method requires fewer communication rounds to reach target accuracy compared to baselines, making it highly suitable for resource-constrained federated environments.

---

> ### Author Response · Authors · 2025-11-26
> **Hyperparameter Specification (Response to W4)**
>
> In the revised manuscript, we have added a comprehensive table in **Appendix A.3** that explicitly lists the values for all key hyperparameters used in our experiments.

---

> ### Author Response · Authors · 2025-11-26
> **Evaluation on Diverse Datasets (Response to W6)**
>
> **Generalization across Domain Shifts (Digits-DG).**
> To demonstrate robustness against feature distribution shifts, we conducted experiments on the **Digits-DG benchmark** [1] (including MNIST [2], SVHN [3], USPS [4]). We set up the experiments with $N=6$ clients, where each domain is assigned to two clients. This simulates a realistic scenario with severe feature distribution shifts. The results below show that FedPAC achieves the highest average accuracy, proving its capability to handle severe feature shifts.
>
> *Table 1: Linear probing accuracy on Digits-DG.*
> | Method | MNIST | SVHN | USPS | AVG |
> | :---: | :---: | :---: | :---: | :---: |
> | BYOL | 96.93 | 79.77 | 94.31 | 90.34 |
> | FedX | 97.77  | 83.09 | 95.56 | 92.14 |
> | FedU2-BYOL | 99.01 | **87.25** | 96.51 | 94.26 |
> | SSD | 98.67 | 86.18 | 96.92 | 93.92 |
> | **FedPAC** | **99.13** | 86.24 | **97.46** | **94.28** |
>
>
> *Table 2: 1% fine-tuning accuracy on Digits-DG.*
> | Method | MNIST | SVHN | USPS | AVG |
> | :---: | :---: | :---: | :---: | :---: |
> | BYOL | 94.78 | 75.98 | 85.07 | 85.28 |
> | FedX | 95.33 | 77.82 | 88.99 | 87.38 |
> | FedU2-BYOL | 97.33 | **81.48** | 92.32 | 90.38 |
> | SSD | 97.57 | 80.29 | 89.65 | 89.17 |
> | **FedPAC** | **97.87** | 79.86 | **93.57** | **90.43** |
>
>
>
> *Table 3: 10% fine-tuning accuracy on Digits-DG.*
> | Method | MNIST | SVHN | USPS | AVG |
> | :---: | :---: | :---: | :---: | :---: |
> | BYOL | 97.96 | 85.77 | 94.01 | 92.58 |
> | FedX | 98.84 | 87.49 | 95.02 | 93.78 |
> | FedU2-BYOL | 98.87 | 89.56 | 96.41 | 94.95 |
> | SSD | 99.01 | **89.91** | 95.31 | 94.74 |
> | **FedPAC** | **99.10** | 89.78 | **96.86** | **95.25** |
>
> **Scalability to Semantic Complexity (Tiny-ImageNet).**
> To assess scalability to datasets with higher resolution and more categories, we evaluated FedPAC on **Tiny-ImageNet** [5]. We adopted a Cross-Silo setting with $N=10$ clients under a non-IID partition ($\alpha=0.1$). As shown below, FedPAC consistently outperforms baselines on Tiny-ImageNet, with a more pronounced performance gap than in CIFAR benchmarks, highlighting its advantage in handling more complex datasets.
>
> *Table 4: Performance comparison on Tiny-ImageNet.*
> | Method | Linear | 1% | 10% |
> | :---: | :---: | :---: | :---: |
> | BYOL | 36.78 | 11.20 | 27.51 |
> | FedX | 39.50 | 13.37 | 32.71 |
> | FedU2-BYOL | 43.74 | 15.65 | 34.22 |
> | SSD | 42.42 | 13.78| 32.84 |
> | **FedPAC** | **47.44** | **20.28** | **38.20** |
>
> These results confirm that FedPAC's superiority is robust and not limited to CIFAR-style datasets.
>
> [1] Ya Li, Xinmei Tian, Mingming Gong, Yajing Liu, Tongliang Liu, Kun Zhang, and Dacheng Tao. Deep domain generalization via conditional invariant adversarial networks. In Proceedings of the European conference on computer vision (ECCV), pp. 624–639, 2018.
>
> [2] Yann LeCun, L´eon Bottou, Yoshua Bengio, and Patrick Haffner. Gradient-based learning applied to document recognition. Proceedings of the IEEE, 86(11):2278–2324, 2002.
>
> [3] Yuval Netzer, Tao Wang, Adam Coates, Alessandro Bissacco, Baolin Wu, Andrew Y Ng, et al. Reading digits in natural images with unsupervised feature learning. In NIPS workshop on deep learning and unsupervised feature learning, volume 2011, pp. 7. Granada, 2011.
>
> [4] Jonathan J. Hull. A database for handwritten text recognition research. IEEE Transactions on pattern analysis and machine intelligence, 16(5):550–554, 2002.
>
> [5] Jia Deng, Wei Dong, Richard Socher, Li-Jia Li, Kai Li, and Li Fei-Fei. Imagenet: A large-scale hierarchical image database. In 2009 IEEE conference on computer vision and pattern recognition, pp. 248–255. Ieee, 2009.

---

> ### Author Response · Authors · 2025-11-30
> **Additional Evaluation on Diverse Datasets (Response to W6)**
>
> We have newly extended our evaluation to DomainNet [1], a large-scale dataset characterized by high semantic complexity and severe distribution shifts. Specifically, we constructed a subset by selecting 10 common categories across six domains following existing works [2,3]. As reported in the table below, we observe significant performance variance among baseline methods across different domains. Methods that excel in specific domains often struggle in others, suggesting they may overfit to specific domain styles rather than learning generalizable features. In contrast, FedPAC demonstrates superior stability and robustness. It ranks within the top-2 in five out of the six domains and achieves the highest average accuracy. These additional results confirm that our method effectively scales to more challenging, realistic scenarios involving diverse data sources, further validating its strong generalization capability.
>
>
> *Table 1: Linear probing accuracy on DomainNet-10.*
> | Method | Clipart | Infograph | Painting | Quickdraw | Real | Sketch | AVG |
> | :---: | :---: | :---: | :---: | :---: | :---: | :---: | :---: |
> | BYOL | 63.64 | 34.45 | 50.59 | 67.79 | 79.33 | 66.25 | 60.34 |
> | FedX | 68.69 | 32.31 | 57.14 | 73.87 | 83.34 | 71.88 | 64.55 |
> | FedU2-BYOL | 67.83 | 35.25 | 61.26 | 76.34 | **86.10** | **74.03** | 66.80 |
> | SSD | 66.95 | **44.01** | **69.75** | 82.53| 69.10 | 69.97 | 67.05 |
> | **FedPAC** | **74.19** | 37.06 | 62.62 | **86.47** | 73.43 | 73.12 | **67.82** |
>
> [1] Xingchao Peng, Qinxun Bai, Xide Xia, Zijun Huang, Kate Saenko, and Bo Wang. Moment matching for multi-source domain adaptation. In Proceedings of the IEEE/CVF international conference on computer vision, pp. 1406–1415, 2019.
>
> [2] Fengda Zhang, Kun Kuang, Long Chen, Zhaoyang You, Tao Shen, Jun Xiao, Yin Zhang, Chao Wu, Fei Wu, Yueting Zhuang, et al. Federated unsupervised representation learning. Frontiers of Information Technology & Electronic Engineering, 24(8):1181–1193, 2023.
>
> [3] Haokun Chen, Hang Li, Yao Zhang, Jinhe Bi, Gengyuan Zhang, Yueqi Zhang, Philip Torr, Jindong Gu, Denis Krompass, and Volker Tresp. Fedbip: Heterogeneous one-shot federated learning with personalized latent diffusion models. In Proceedings of the Computer Vision and Pattern Recognition Conference, pp. 30440–30450, 2025.

---

> ### Author Response · Authors · 2025-11-30
> **Summary of Revisions**
>
> We sincerely thank the reviewer for the time and effort dedicated to reviewing our manuscript. In response, we have added:
> - Expanded References and Baselines (**W1, W7, Q4**): Incorporated the latest SOTA method and more recent works (**Section 2.2 & 2.3**).
> - Robustness Analysis (**W2, Q1**): Added experiments on pathological Non-IID settings(**Appendix C.2**).
> - Complexity & Privacy Discussion (**W3, W5, Q2, Q3**): Added theoretical complexity analysis and discussion on privacy concerns (**Appendix A.4**).
> - Hyperparameter Details (**W4**): Provided a comprehensive table of hyperparameter settings (**Appendix A.3**).
> - Broader Datasets (**W6**): Significantly extended our evaluation to include Digits-DG, DomainNet, and Tiny-ImageNet datasets (**Appendix C.2 & C.4**).
>
> We thank you again for your valuable feedback, which has helped improve the quality of our paper.

---

### Meta-Review · Area_Chair_zGAz · 2026-01-07

**Summary:**

The reviewers' concerns are mostly on 1) technical novelty and significance. There are lack of recent works (after 2024) discussed in the paper, and the distinction between current work and prior works are not clear. 2) Insufficient experimental results. Reviewers suggest the paper to provide results on additional datasets, task types, sensitivity analysis and more SOTA methods. 3) Presentation rigor. There are notation conflicts and presentation inconsistencies.

**Reviewer Concerns:**

The authors provided a very detailed rebuttal, and included additional experimental results as requested by the reviewers. Considering the significant length and volume of the revised contents, from new discussions on recent related works and privacy analysis, to new results and discussion on additional datasets/heterogeneity level/tasks, I believe this paper will benefit from a new round of careful review before it can be accepted.

**Reviewer Scores:**

I think the rebuttal quality is high, and all the reviewers may likely to maintain or even raise their scores given that they think the rebuttal content is helpful to address their concerns. There is some possibility that reviewers may raise further questions or concerns on the newly introduced results and discussion.

---

### Decision · Program_Chairs · 2026-01-26

Reject